# Epigenetic and molecular coordination between HDAC2 and SMAD3-SKI regulates essential brain tumour stem cell characteristics

Ravinder K. Bahia[1,2], Xiaoguang Hao[1,2], Rozina Hassam[1,2], Orsolya Cseh[1,2], Danielle A. Bozek[1,2], H. Artee Luchman [1,2,3] ✉ & Samuel Weiss [1,2,3] ✉

Histone deacetylases are important epigenetic regulators that have been reported to play essential roles in cancer stem cell functions and are promising therapeutic targets in many cancers including glioblastoma. However, the functionally relevant roles of specific histone deacetylases, in the maintenance of key self-renewal and growth characteristics of brain tumour stem cell (BTSC) sub-populations of glioblastoma, remain to be fully resolved. Here, using pharmacological inhibition and genetic loss and gain of function approaches, we identify HDAC2 as the most relevant histone deacetylase for re-organization of chromatin accessibility resulting in maintenance of BTSC growth and self-renewal properties. Furthermore, its specific interaction with the transforming growth factor-β pathway related proteins, SMAD3 and SKI, is crucial for the maintenance of tumorigenic potential in BTSCs in vitro and in orthotopic xenograft models. Inhibition of HDAC2 activity and disruption of the coordinated mechanisms regulated by the HDAC2-SMAD3-SKI axis are thus promising therapeutic approaches for targeting BTSCs.

Glioblastoma (GBM) brain tumor stem cells (BTSCs) display cancer stem cell (CSC) properties of high self-renewal, aberrant multi-lineage differentiation, and tumorigenesis; features that create incredible disease complexity leading to therapeutic resistance and recurrence[1–12]. Cancer has been traditionally viewed as a genetic disease in origin, but it is now evident that epigenetic mechanisms play critical roles in the regulation of many cellular functions. BTSCs have been shown to adapt to epigenetic alterations, such as aberrant histone methylation or acetylation patterns induced by the deregulation of relevant epigenetic regulators[1–5,13–15]. These epigenetic changes modify the chromatin structure of BTSCs in a manner that facilitates downstream transcriptional programs associated with their stemness, tumorigenic potential, and drug resistance features[1–5,13–15].

Histone deacetylases (HDACs), epigenetic regulators that function to silence gene expression, are frequently found to be deregulated in many human cancers including GBM[1,16–18]. The four different classes of HDACs function as part of multiprotein corepressor complexes and tightly control DNA accessibility and associated transcriptional programs during embryonic stem cell state and early and late stages of neurodevelopment[19–25]. HDACs regulate diverse epigenetic processes in normal stem cell or primed progenitor cell contexts[22–24]. Similar processes are at play in cancer, and the potential therapeutic benefits of pan-HDAC inhibitors for GBM and other cancers have been duly reported[1,17,18,26,27]. However, pan-HDAC inhibitors come with lack of specificity, leading to a plethora of side effects and toxicity[1,18]. There is thus still a need to identify and validate the most relevant HDAC

[1]Arnie Charbonneau Cancer Institute, Cumming School of Medicine, University of Calgary, Calgary, AB, Canada. [2]Department of Cell Biology and Anatomy, University of Calgary, Calgary, AB, Canada. [3]These authors jointly supervised this work: H. Artee Luchman, Samuel Weiss. ✉e-mail: aluchman@ucalgary.ca; weiss@ucalgary.ca

enzyme/s in different disease contexts and, in turn, enable development of specific HDAC inhibitors. Recently, HDAC1 knockdown was reported as having effects on the glioma stem cell phenotype by inducing p53-dependent mechanisms of cell death[28]. Here, we ask whether different HDACs, in collaboration with their disease-specific critical molecular regulators, have unique epigenetic regulations in BTSCs with different genetic and molecular backgrounds.

In this work, we use a cohort of patient-derived GBM BTSCs, reflecting the molecular and genetic heterogeneity of their primary parent tumors (Table 1)[5,29–32], to investigate the functional relevance of specific HDACs. We establish that HDAC2 is the most relevant epigenetic regulator for maintenance of cancer stem cell characteristics and tumorigenesis in BTSCs of different genetic backgrounds. We report that HDAC2, in collaboration with the specific TGF-β pathway proteins, SMAD3-SKI, regulates chromatin organization and associated transcriptional programs to maintain the stem cell and tumorigenic characteristics of GBM BTSCs. Inhibition of HDAC2 activity with specific inhibitors and disruption of this interaction thus provide attractive therapeutic strategies for GBM.

## Results

### HDAC1 and HDAC2 are required for the BTSC growth and for the maintenance of stem cell characteristics

We asked whether HDACs, which have been reported as having roles in normal development and cancers[1,16–24], may also regulate the CSC properties of BTSCs. We first assessed the protein levels of class I (HDAC1, 2, and 3) and II (HDAC6 and 9) HDACs in BTSC lines representing different genetic backgrounds (Table 1), normal human induced pluripotent stem cells (hiPSCs), normal neural stem cells (NSCs), astrocytes derived from the hiPSCs and astrocytes derived from normal human fetal neural stem cells (HF NSCs). We found that class I HDACs were highly expressed in the BTSC lines tested, compared to astrocytes derived from both hiPSCs or HF NSCs (Fig. 1a, b). The expression of HDAC1 and 2 (HDAC1/2) was comparable in hiPSCs, HF NSCs and in the BTSC lines. HDAC2 expression was also high in NSCs derived from hiPSCs but lower in their differentiated astrocyte derivatives, as previously reported in the context of normal neurodevelopment[24]. These findings led us to hypothesize that, like normal iPSCs and NSCs, the high protein levels of HDAC1/2 may indicate their functionally relevant roles for the maintenance of stem cell features and growth of GBM BTSCs.

To further determine the functional relevance of class I HDACs in BTSCs, in comparison to the other classes, pan-HDAC (panobinostat and pracinostat) and class I and II specific HDAC inhibitors (4SC202, mocetinostat, RGFP966, romidepsin, and WT161) were tested in 9 BTSC lines and normal human fetal astrocytes (HFAs). We found that the HDAC1/2 specific inhibitor, romidepsin, reduced BTSC viability, at lower concentrations (300–1000 pM) than the pan-HDAC inhibitors and other specific HDAC inhibitors (Fig. 1c; Supplementary Fig. 1a–f).

HDAC1/2 have been shown to be essential for cell cycle progression and stem cell self-renewal in human embryonic stem cells (hESCs)[22]. We asked whether inhibition of HDAC1/2 in BTSCs would induce any changes in survival, cell cycle progression, and self-renewal. Treatment of BTSC lines with sub-IC50 concentrations of 50pM and 100pM romidepsin for 5 days followed by Annexin V/7AAD staining and flow cytometric analysis showed a greater increase in the frequency of Annexin V+ cells (dying cells) at higher doses of romidepsin (Supplementary Fig. 2a, b; Supplementary Fig. 3a, b). EdU cell cycle flow cytometric analysis further showed a significant decrease in the percentage of cells in S phase in romidepsin (50pM) treated BTSCs compared to vehicle treated cells (Supplementary Fig. 2c, d; Supplementary Fig. 3c, d). This was concomitant with a significant increase in the percentage of G2/M phase cells, indicative of cell cycle arrest (Supplementary Fig. 2c, d; Supplementary Fig. 3c, d). Limiting dilution assays (LDAs) showed that BTSC self-renewal was significantly decreased at a low picomolar dose (50pM) of romidepsin (Fig. 1d; Supplementary Fig. 3e). We also observed a striking change in the morphology of BTSC lines following treatment with 50pM romidepsin (Supplementary Fig. 2e; Supplementary Fig. 3f); suggesting that the cells may be undergoing cell-fate specification programs. Collectively, these results suggest that inhibition of HDAC1/2 is critical for the regulation of BTSC survival, cell cycle progression and stem cell characteristics.

### Global gene expression analysis following HDAC1/2 inhibition with romidepsin reveals a modulation in components of the TGF-β pathway

HDAC enzymes catalyze the deacetylation of lysine residues of histone H3 and H4 and directly regulate the expression of 5–20% of all genes[1,19]. Specifically, HDAC1/2 regulate the acetylation of H4 histone at lysine 5, 12, 16, and 20 at regulatory regions[33,34] and H3K18 crotonylation (a histone mark of open chromatin) levels at transcription start sites of the target genes in a particular cellular context[35]. The remaining indirect effects of HDAC enzymes are mediated though deacetylation of non-histone proteins and the epigenetic modulation of downstream signaling pathways[19,36–38].

We evaluated the effect of pan- and specific-HDAC inhibitors on global acetylation patterns and any associated impacts on gene regulation. Interestingly, treatment with romidepsin for 72 h increased the acetylation at H3 and H4 lysine residues in BTSCs to the levels observed with pan and other class I and II HDAC inhibitors (Fig. 1e; Supplementary Fig. 4a, b, e). Specific inhibition of HDAC1/2 with romidepsin increased acetylation of H3 at lysine 9, 18, 27, 56 and of H4 at lysine 5, 12, 16, and 20 in BTSC lines (Fig. 1e; Supplementary Fig. 4a, b). An increase in the crotonylation of H3 lysine at 18 was also observed following romidepsin treatment compared to vehicle control (Fig. 1e; Supplementary Fig. 4a). These data suggest that HDAC1/2 contribute to changes in global acetylation and

**Table 1 | BTSC lines used for drug screening representing genetic variability**

| BTSC ID | Age | Sex | Diagnosis | MGMT status | EGFR status | p53 status | PTEN status | IDH1/2 status |
|---------|-----|-----|-----------|-------------|-------------|------------|-------------|---------------|
| BT48 | 68 | Male | GBM | M | mt | wt | het | wt/wt |
| BT50 | 61 | Male | GBM | um | wt | wt | het | wt/wt |
| BT67 | 44 | Male | GBM | M | wt | wt | het | wt/wt |
| BT69 | 50 | Male | GBM/GS | um | het | wt | mt | wt/wt |
| BT89 | 59 | Female | GBM | M | wt | wt | wt | wt/wt |
| BT94 | 60 | Female | GBM | M | wt | wt | mt | wt/wt |
| BT147 | 56 | Male | GBM-r | U | vIII | mt | mt | Wt/wt |
| BT189 | 55 | Female | GBM | U | wt | mt | wt | wt/wt |
| BT248Z | 56 | Female | GBM | M | wt | mt | wt | wt/wt |

*GBM-r* recurrent GBM, *um* hemimethylated, *U* unmethylated, *M* methylated, *mt* mutated, *wt* wildtype, *het* heterozygous.

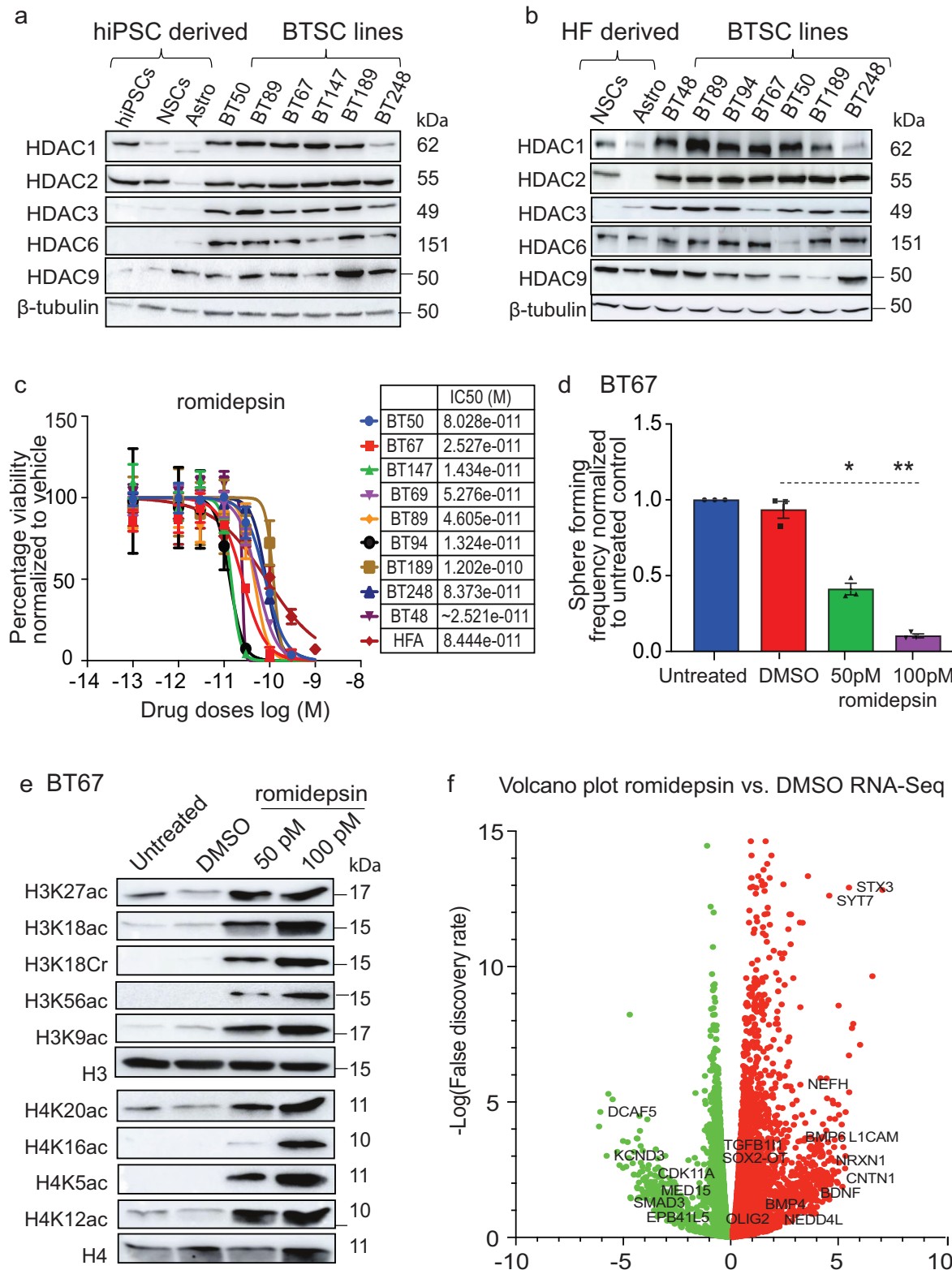

crotonylation levels, typically associated with active chromatin status and increased transcriptional levels, in BTSCs.

We next performed RNA-sequencing to investigate the impact of global acetylation and crotonylation changes on associated gene expression levels, after 72 h romidepsin (50pM) treatment, in the BT67 line relative to DMSO vehicle-treated cells. Genes related to neurogenesis and synaptogenesis (*BDNF, CNTN1, NEFH, NRXN1, SYT7,* and

*STX3*) [21–24,39] and to neural stem and progenitor cell fates (*SOX2* and *OLIG2*)[40] were amongst the differentially expressed genes in romidepsin treated vs. vehicle-treated BT67 cells (Fig. 1f, Supplementary data 1). Gene set enrichment analyses (GSEAs) showed enrichment of the downregulated genes in the gene sets associated with cell cycle regulation, DNA-damage response and axon guidance and of the upregulated genes in neuronal and synaptic functions and the

**Fig. 1 | HDAC1 and 2 are required for BTSC growth and stem cell characteristics by regulating associated histone modifications and transcriptional programs.** Protein levels of different classes of HDACs in BTSC cell lines (**a**) relative to hiPSCs and their normal NSCs and astrocyte derivatives (**b**) relative to normal HFs and HFAs. **c** Cell viability for 9 BTSC lines and HFAs from three biological replicates following 14 days treatment with HDAC1/2 specific inhibitor romidepsin (100–1000 pM) Data represent mean values ± SD, $n = 3$. **d** Sphere forming frequency of BT67 cells following 21 days treatment with 50 pM and 100 pM doses of romidepsin as assessed with limiting dilution assays. Significance was determined using ANOVA (Dunnett's test) at 95% confidence intervals, *$p < 0.05$, **$p < 0.01$, Data represent mean ± upper and lower 95% confidence intervals, $n = 3$. **e** Acetylation levels of lysine residues of histones, H3 and H4, and crotonylation of H3 at lysine 18 in BT67 cells following 72 h treatment with romidepsin (50 pM and 100 pM) and vehicle control. **f** Volcano plot depicting the differential expression of transcripts associated with stem cell functions (*SOX2* and *OLIG2*), neuronal differentiation (*CNTN1, BDNF, NRXN1, SYT7*, and *NEFs*) and a modulation of transcripts associated with the TGF-β pathway such as *SMAD3, TGFB1l1, SKI, NEDD4L, BMP4*, and *6* in romidepsin treated BT67 cells relative to vehicle control. Green dots denote transcripts that are significantly downregulated and red dots denote significantly upregulated transcripts in vehicle treated vs. romidepsin treated BT67 samples. *$p < 0.05$ for all genes. $n = 3$ biologically independent. Brain tumor stem cells (BTSCs), human induced pluripotent stem cells (hiPSCs), neural stem cells (NSCs), human fetal stem cells (HFs), human fetal astrocytes (HFAs)" represents molecular weight markers (50, 37, 15, and 10). Source data are provided in the source data file.

G-protein couple receptor signaling related gene sets in response to romidepsin treatment (Supplementary Fig. 5a–e). We further validated these expression changes at mRNA and protein levels and confirmed that romidepsin treatment leads to the downregulation of SOX2 and OLIG2 and upregulation of BDNF, GFAP and STX3 (Fig. 2a; Supplementary Fig. 4c; Supplementary Fig. 6a–c, 6g–l; Supplementary Fig. 7a–d). These changes were associated with an increase in the protein levels of cell cycle regulators, p21 and p38, which are known to induce cell cycle arrest (Fig. 2a; Supplementary Fig. 4c). These data further support the changes observed in BTSC self-renewal, cell morphology and cell cycle progression following HDAC1/2 inhibition.

Interestingly, the RNA-seq data also showed modulation in the expression levels of transcripts associated with the TGF-β pathway such as *SMAD1, SMAD3, SKI, BMP4* and *BMP6* and *NEDD4L* in BT67 cells treated with romidepsin relative to DMSO vehicle-treated cells (Fig. 1f; Supplementary Fig. 5e). Decreases in the protein levels of total and phospho-SMAD3 and SKI and an increase in the levels of NEDD4L and the inhibitory SMAD protein, SMAD7, were also confirmed by western blot in romidepsin vs. vehicle-treated samples (Fig. 2b; Supplementary Fig. 4d; Supplementary Fig. 6d–f, 6j–l). Furthermore, an increase in the levels of ubiquitinated-SMAD3 was observed following treatment with romidepsin (Supplementary Fig. 5f). The above data suggest that an increase in the mRNA and protein abundance of NEDD4L, following romidepsin treatment (Fig. 2b; Supplementary Fig. 4d), may contribute to ubiquitin-mediated degradation and related decrease in the total and phospho-SMAD3 proteins in BTSCs, as previously reported in the context of mouse ESCs[41]. Interestingly, this modulation in the components of the TGF-β pathway was specific to the inhibition of HDAC1/2 with romidepsin and was not observed with any other HDAC inhibitors (Supplementary Fig. 4f). These data suggest that, together, HDAC1/2 epigenetically regulate transcriptional programs, including the epigenetic modulation of the TGF-β pathway, that support BTSC growth and stemness.

## HDAC1/2-mediated histone deacetylation govern the transcriptional state of genes related to self-renewal, cell-fate, and TGF-β signaling in BTSCs

We next performed H3K27ac ChIP-sequencing to assess changes in H3K27ac status at 5′ regulatory regions of differentially expressed genes related to BTSC stemness, differentiation, and the TGF-β pathway-related genes following romidepsin treatment. A reduction in the levels of H3K27ac at the 5′ regulatory regions of the stem cell gene, *SOX2*, and the TGF-β pathway-related genes, *SMAD3* and *SKI* was observed in the romidepsin treated cells (Fig. 2c–f). A small increase in H3K27ac levels at the 5′ regulatory region of the *BDNF* gene was also observed (Fig. 2f). These changes in the levels of H3K27ac were validated by ChIP-qPCR in romidepsin treated vs. vehicle-treated BTSC samples (Fig. 2g, h; Supplementary Fig. 7e, f).

Since there was little change observed in H3K27ac levels at the 5′ regulatory end of the *BDNF* gene, we next asked whether HDAC1/2 regulated acetylation of H4 lysine could also have contributed to the observed gene expression changes. Using H4K5ac ChIP-PCR, we indeed found a significant upregulation in the levels of H4K5ac at the *BDNF* 5′ end with romidepsin treatment (Fig. 2j), as previously reported in a traumatic brain injury model[34]. We further found that romidepsin treatment led to a significant reduction in the H4K5ac levels at the 5′ region of the *SMAD3* gene (Fig. 2i) which may, in part, be responsible for its transcriptional repression (Fig. 1f; Supplementary Fig. 7a, c), with no significant changes at the 5′ region of the *SOX2* and *SKI* genes (Supplementary Fig. 7g, h). Taken together, these data confirm that inhibition of HDAC1/2 results in reorganization of global histone acetylation patterns, thus inducing changes in BTSC transcriptional programs that are important for their self-renewal and that may contribute to their differentiation abilities.

## HDAC2 has more diverse roles than HDAC1 in regulating BTSC growth and specific histone acetylation and transcriptional programs

We next asked which of HDAC1 or HDAC2 was functionally more relevant in regulation of these BTSC features. Recently, HDAC1 knockdown was reported as having effects on the glioma stem cell phenotype by inducing p53-dependent mechanisms of cell death[28]. Here, we used CRISPR-Cas9 technology with two different guide RNAs to target *HDAC1/2* genomic loci along with an AAVS1 targeted control gRNA. Both guide RNAs were equally effective at reducing HDAC1/2 protein levels in BTSCs relative to AAVS1 control (Fig. 3a–c; Supplementary Fig. 8a–c). Further assessment of single and double KO of *HDAC1/2* on BTSCs viability revealed that while *HDAC1* KO reduced BTSC viability (~40%) significantly (Fig. 3d; Supplementary Fig. 8d), the loss of cell viability was even more dramatic (~60%) in *HDAC2* KO BTSCs (Fig. 3d; Supplementary Fig. 8d; Supplementary Fig. 9a, c) compared to the AAVS1 control BTSCs. *HDAC2* KO significantly decreased the sphere-forming frequency of BTSCs compared to *HDAC1* KO and AAVS1 control (Fig. 3e; Supplementary Fig. 8e). Furthermore, there was a greater reduction in the percentage of S phase cells with *HDAC2* KO (Fig. 3f; Supplementary Fig. 8f; Supplementary Fig. 9b, d) compared to the AAVS1 control and *HDAC1* KO cells. Strikingly, a significant increase in the percentage of cells arrested in G2/M phase of the cell cycle was specific to the *HDAC2* KO BTSCs (Fig. 3f; Supplementary Fig. 8f; Supplementary Fig. 9b, d). Overall, these data suggest that HDAC2 is more critical than HDAC1 for the maintenance of BTSC viability, stemness and cell cycle progression in vitro.

We next investigated the epigenetic mechanisms through which HDAC1/2 regulate BTSC growth. We observed an increase in H3 and H4 lysine acetylation levels in BTSCs with KOs targeting either or both *HDAC1/2* loci compared to the AAVS1 controls (Fig. 3g; Supplementary Fig. 8g). Interestingly, increase in global H4K5ac and H3K18Cr levels were higher in the *HDAC2* KOs than the *HDAC1* KOs (Fig. 3g; Supplementary Fig. 8g). Further investigation of the functional relevance of HDAC2 specific histone modifications in BTSCs using H4K5ac ChIP-seq with AAVS1 and *HDAC2* KO BT67 cells showed a drop in the enrichment levels of H4K5ac at 5′ regulatory regions of *SOX2, OLIG1/2, SMAD3,* and *SKI* genes. In addition, a gain of broader domains of H4K5ac at the regulatory regions of differentiation markers such as *BDNF, L1CAM,*

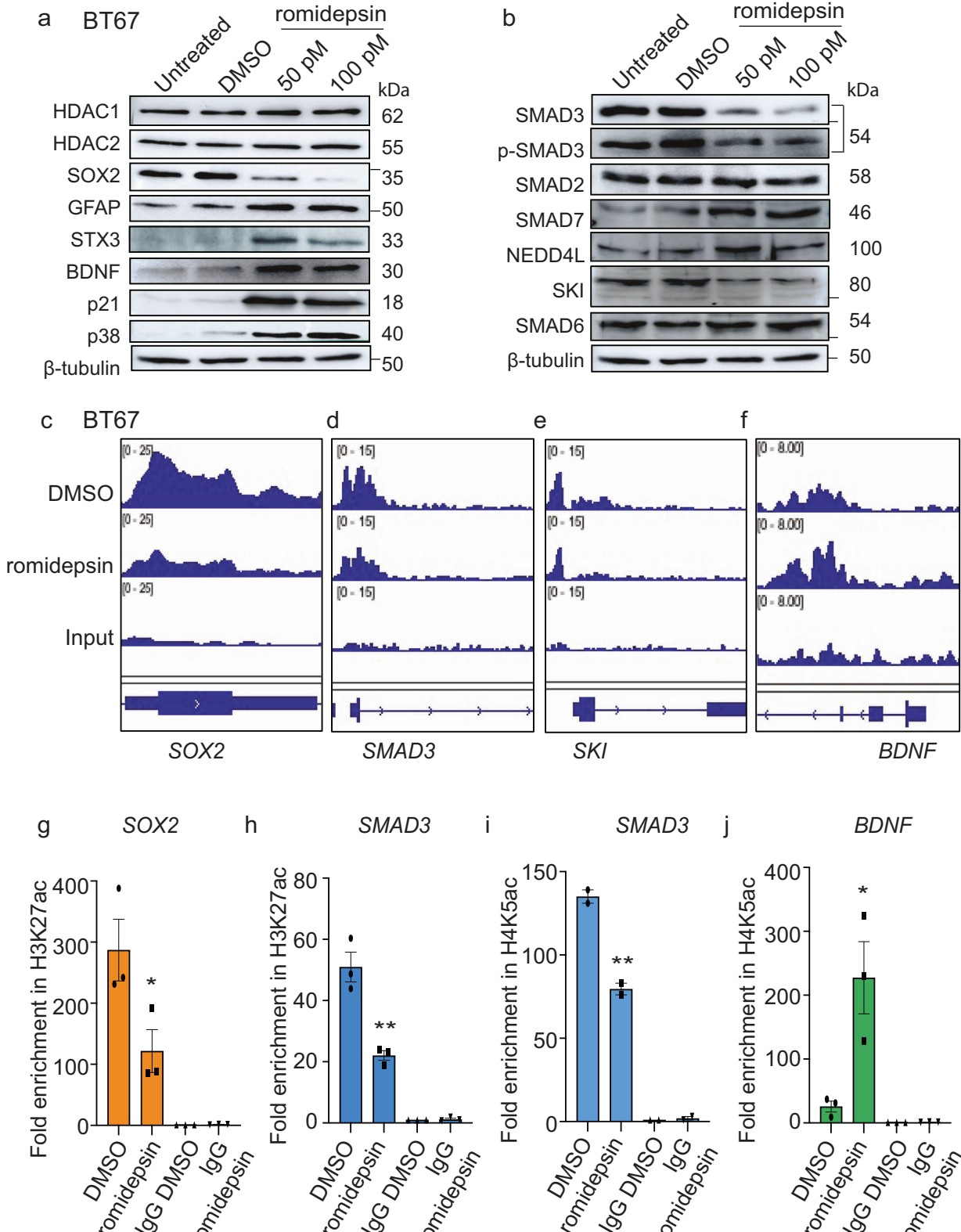

*NEUROD1/2, NEFH* and *STX3* (Fig. 3h–m; Supplementary Fig. 10a–e) was observed, and validated by ChIP-qPCR (Supplementary Fig. 10f–m). Similarly, the enrichment levels of H3K27ac were reduced at the 5' region of *SOX2, SMAD3* and *SKI* with an increase at the 5' end of *BDNF* genes, particularly, in *HDAC2* KO BTSCs (Supplementary Fig. 11a–h). Further examination of the impact of single and double *HDAC1/2* KOs showed a reduction in protein levels of SOX2 and OLIG2 (Fig. 3a–c;

Supplementary Fig. 8a–c), with a more dramatic impact on SOX2 protein levels with the loss of HDAC2 compared to HDAC1 (Supplementary Fig. 12a, b, e, f). These changes were accompanied with an increase in BDNF, NEDD4L, and SMAD7 protein levels and a reduction in total- and phospho-SMAD3 and SKI proteins was only seen when *HDAC2* was targeted (Fig. 3a–c; Supplementary Fig. 8a–c; Supplementary Fig. 12c, d, g, h). This increase in BDNF expression with the

**Fig. 2 | HDAC1/2 mediated histone deacetylations govern the transcriptional state of genes related to self-renewal, cell-fate, and TGF-β signaling in BTSCs.** **a** Validation of RNA-seq data confirming the changes in the protein levels of SOX2, GFAP, STX3 and BDNF, and the cell cycle regulators, CDKN1A (p21) and p38 in BT67 vehicle treated vs romidepsin treated cells for 72 h $n = 3$. **b** Western blot validation of components of TGF-β pathway including total and phospho-SMAD3, negative regulator, proto-oncogene, SKI, inhibitory SMAD protein, SMAD7 and negative regulator of SMAD3, NEDD4L. $n = 3$. **c**–**f** Changes in the levels of the H3K27ac mark at the 5' regulatory region of *SOX2*, *BDNF* and TGF-β pathway-related genes, *SMAD3*,

*SKI*. **g**, **h** Validation of ChIP-seq data by ChIP-PCR. Significance was determined using unpaired two-tailed t-test with 95% confidence intervals, *$p < 0.05$, **$p < 0.0048$; data are represented as fold enrichment mean values ± SEM; $n = 3$. **i**, **j** Assessment of changes in the H4K5ac mark by ChIP-PCR at the 5' region of *SMAD3* and *BDNF* genes. Significance was determined using unpaired two-tailed t-test with 95% confidence intervals, *$p < 0.024$, **$p < 0.009$; data are represented as fold enrichment mean values ± SEM; $n = 3$). - represents molecular weight markers (50 and 37). Source data are provided in the source data file.

*HDAC2* KO is possibly a consequence of the concomitant increase in H4K5ac levels[34]. These data lead us to propose that HDAC2 modulates BTSC chromatin architecture by regulating specific histone acetylation patterns and by modulating the components of the TGF-β pathway, thus display more diverse functional regulations in determining BTSCs' cell-fate specification versus self-renewal properties.

### Loss of HDAC2 improves median survival in orthotopic xenograft BTSC models

We next asked if genetic deletion of either HDAC1 or 2 or of both would show similar effects on tumor cell growth in vivo as those observed in vitro. Mice orthotopically xenografted with *HDAC1* KO p53-wildtype BT67 cells showed an improvement in median survival ($p < 0.0008$, Fig. 4a). Mice orthotopically xenografted with BT67 *HDAC2* KOs cells demonstrated an even greater improvement in median survival than the AAVS1 control and *HDAC1* KO xenografts (Fig. 4a). We further confirmed these findings using shRNA mediated knockdown approaches in two different p53-wildtype (BT67) and p53-mutant (BT147) BTSC cultures (Supplementary Fig. 13a–c). Similar to the CRISPR-cas9 KO studies, we found that HDAC2 KD resulted in a significant reduction in BTSCs cell viability in vitro (Supplementary Fig. 13d, f). Furthermore, a significant increase in median survival of mice orthotopically xenografted with HDAC2 KD BT147 and BT67 cells, irrespective of p53-status, compared to their respective scrambled control mice was observed (Fig. 4b; Supplementary Fig. 13e). In contrast, there was no apparent benefit in overall survival in mice orthotopically xenografted with HDAC1 KD BT147 relative to the scrambled control mice (Supplementary Fig. 13g); consistent with the previous study on the role of HDAC1 in p53-wildtype BTSCs[28]. These findings further confirm that HDAC2 is more relevant of the two HDACs for BTSC growth both in vitro and in vivo.

### HDAC2 interacts with the TGF -β related proteins, SMAD3-SKI, in BTSCs

HDAC1/2 have been shown to be part of diverse multiprotein complexes including their complex with the TGF-β pathway-related proteins[19,20,25,42]. Our findings above suggest that inhibition of HDAC2 results in decrease in mRNA and protein abundance of SMAD3 and SKI. We therefore asked whether the transcription factor, SMAD3, along with the cofactors, SKI or SNON, enables the recruitment of HDAC2 in a manner that influences BTSC chromatin landscape and transcriptional programs. Using HDAC1, HDAC2 and SMAD3 antibodies in co-immunoprecipitation (co-IP) assays, we identified protein-protein binding of HDAC2, but not of HDAC1, with SMAD3 and SKI proteins (Fig. 4c–f). We did not observe any interaction of HDAC2 with SMAD2 and other negative regulator proteins such as SNON (Fig. 4c–f). To further confirm these findings, we treated BTSCs with romidepsin or vehicle control for 48 h. We observed a reduction in the levels of immunoprecipitated total and phospho-SMAD3 and SKI proteins when the HDAC2 antibody was used for immunoprecipitation in lysates from BTSCs treated with romidepsin vs vehicle-treated samples (Supplementary Fig. 14b). Conversely, a reduced pull down of HDAC2 protein was observed following immunoprecipitation with SMAD3 antibody, in romidepsin treated BTSC lysates compared to vehicle-treated samples (Supplementary Fig. 14c). We also detected pull down of SMAD3 and

HDAC2 proteins with SKI and SMAD4 antibodies (Supplementary Fig. 14d, e), suggesting their association with the SMAD3 and HDAC2 protein complex.

Next, to confirm whether the destabilization of HDAC2 and SMAD3-SKI interaction is due to a decrease in the levels of SMAD3 protein following HDAC2 inhibition, we performed time-course treatments with romidepsin for 24, 48, 72, and 96 h. We observed dramatic decreases in the protein levels of total and phospho-SMAD3 48 h post-treatment with associated increases in NEDD4L and SMAD7 proteins (Supplementary Fig. 14f). However, a decrease in SMAD3-SKI immunoprecipitated proteins with the HDAC2 antibody was observed 24 h post-treatment with romidepsin relative to the vehicle control (Supplementary Fig. 14g). We further found a decrease in the pull-down efficiency of HDAC2 antibody when the cells were treated for 72 h with romidepsin relative to the vehicle control. We postulate that this may, in part, be attributed to changes in the cell state or disruption of protein complex impacting the crosslinking and/or immunoprecipitation efficiencies (Supplementary Fig. 14g; Supplementary Fig. 16a–c; Supplementary Fig. 17a–c).

We further confirmed these findings with in situ proximity ligation assays (PLAs). The results showed a significantly higher number of PLA foci per nuclei in anti-HDAC2+anti-SMAD3, anti-HDAC2+anti-SKI, and anti-SMAD3+anti-SKI compared to anti-HDAC1+anti-SMAD3 samples which were reduced following treatment with romidepsin for 24 h relative to the vehicle control (Fig. 4g; Supplementary Fig. 15; Supplementary Fig. 16d, e). These data suggest that HDAC2 may be a binding partner of SMAD3-SKI in BTSCs.

HDAC2 and SMAD3 proteins each have different functional domains for the regulation of diverse biological functions[43,44]. The MH1 domain of the SMAD3 protein has been associated with HDACs' enzymatic activity and regulation of transcriptional programs[44]. We thus mapped the binding domains of HDAC2 in SMAD3 protein or vice versa by introducing flag-tagged mutant clones of SMAD3 or HDAC2 proteins in HEK293T/17 cells (Supplementary Fig. 14h–j, Supplementary data 6). Co-IP assays with an anti-flag antibody showed increased pull down of HDAC2 protein in lysates from HEK293T/17 cells with clones expressing MH1 domain, particularly, SMAD3NL, relative to the SMAD3C or empty vector expressing cells (Fig. 4h). Similarly, a HDAC2N clone, expressing the HDAC association domain, showed increased binding with SMAD3 relative to the lysates from HEK293T/17 cells expressing either HDAC2C clone or empty vector (Fig. 4i). These findings are in line with the studies above and suggest that the MH1 domain of SMAD3 protein may be critical for protein-protein interactions with the HDAC association domain of HDAC2. Disruption of this epigenetic and molecular coordination may, in turn, be a key event leading to the changes in downstream cellular and molecular programs of BTSCs.

### HDAC2 and SMAD3 co-regulate the self-renewal versus cell-fate specification potentials of BTSCs

We next performed ChIP-sequencing to understand the cooperative mechanisms regulated by HDAC2 and SMAD3-SKI and identify common target genes. ChIP-seq data revealed that a greater overlap between HDAC2 and SMAD3 binding peaks within BTSC genome than HDAC1 (Fig. 5a, Supplementary data 2–4). Further motif enrichment

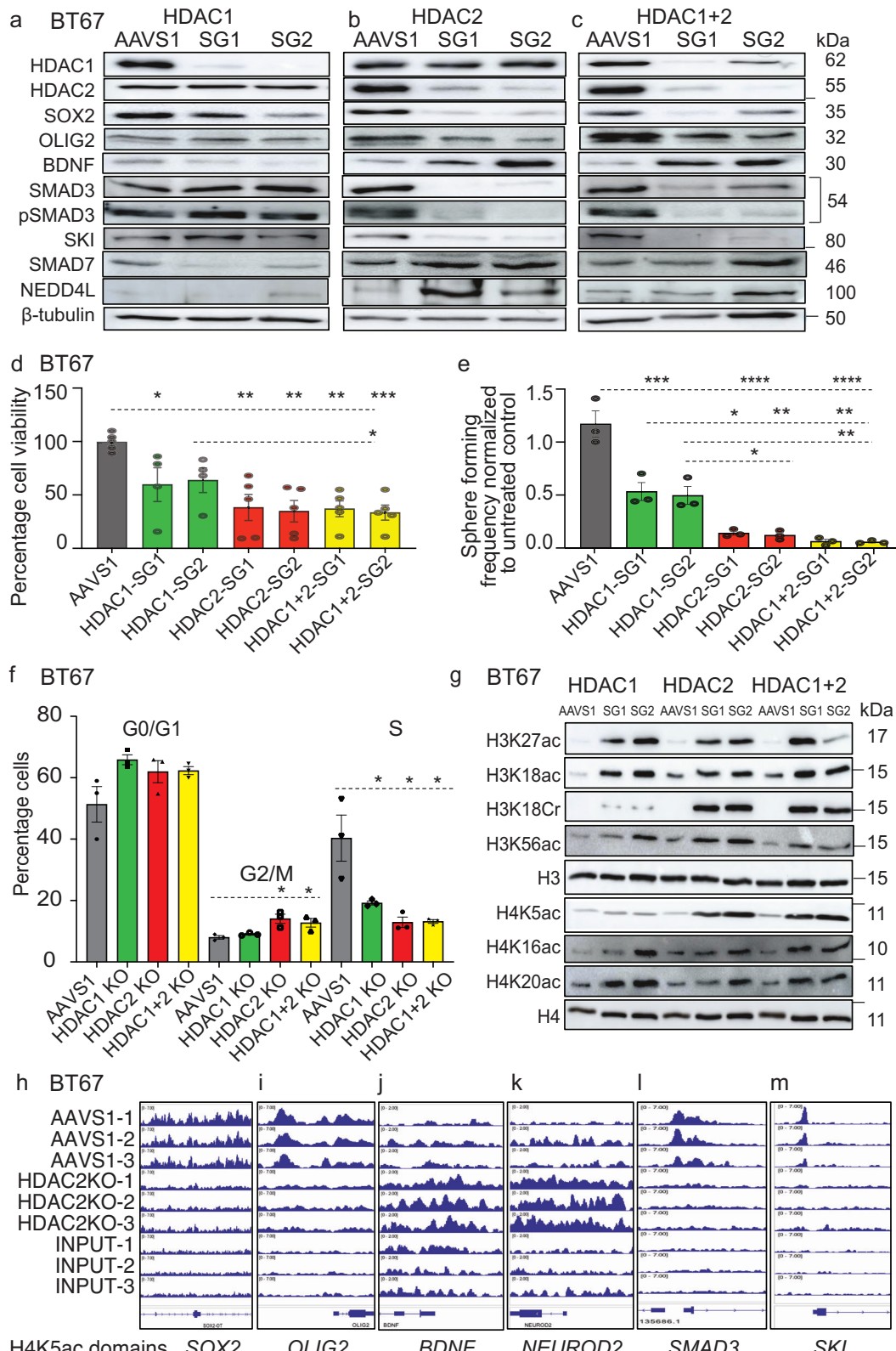

analysis of HDAC2 and SMAD3 overlapping peaks revealed higher percentage of overlapping sequences enriched with *SMAD3, NEUROD1* and *SOX* family related binding motifs (Fig. 5b, Supplementary data 5), suggesting that the genes exhibiting these motifs are coregulated by HDAC2 and SMAD in BTSCs. The differentially expressed transcripts from our RNA-seq data included *BDNF, L1CAM, NCAM1*; cell cycle genes such as *CDKN1A/p21*; and TGF-β pathway-related genes including

*NEDD4L* and *SMAD7* which showed enrichment of both HDAC2 and SMAD3 within their genomic regions (Fig. 5d, f, g; Supplementary Fig. 17a, c). In contrast, the 5' regulatory regions of *SOX2* and *SKI* showed increased binding of SMAD3 whereas HDAC2 had increased binding at the genomic region of *L1CAM* gene (Fig. 5c, e; Supplementary Fig. 17b). We further validated these results with ChIP-qPCR (Fig. 5h–k; Supplementary Fig. 17d–g). These results suggest that

**Fig. 3 | HDAC2 has more diverse roles than HDAC1 in regulating BTSC growth and specific histone modifications and transcriptional programs.**
**a−c** Validation of single and double CRISPR-cas9 mediated *HDAC1/2* KOs in BT67 cells using two independent guide RNAs for each gene. AAVS1 was used as CRISPR-cas9 cut control. Western blots showing changes in protein levels of stem cell regulators; SOX2 and OLIG2, neuronal fate-specific marker; BDNF and the TGF-β pathway-related proteins including total and phospho-SMAD3, SKI, SMAD7, and NEED4L in single and double *HDAC1/2* KO cells relative to the AAVS1 control cells. **d** Effect of single and double *HDAC1/2* KOs on the cell viability of BT67 cells relative to AAVS1 control cells. Significance was determined using ANOVA (Tukey's test) at 95% confidence intervals, *$p < 0.05$, **$p < 0.0021$, ***$p < 0.0009$; data are represented as mean values ± SEM, ($n = 3$, with two independent gRNAs. **e** Sphere forming frequency of BT67 cells following single and double knockout of *HDAC1*

and *2* in BT67 cells relative to AAVS1 cut control. Significance was determined using ANOVA (Tukey's test) at 95% confidence intervals, *$p < 0.05$, **$p < 0.01$, ***$p < 0.001$, ****$p < 0.000$; Data represent mean ± upper and lower 95% confidence intervals, $n = 3$. **f** EdU assays showing the effect of single and double *HDAC1/2* KOs on different phases of cell cycle. Significance was determined using ANOVA (Dunnett's test) at 95% confidence intervals, *$p < 0.048$, **$p < 0.009$; data are represented as mean values ± SEM; $n = 3$ Gating strategies are provided in Supp. Fig. 22a. **g** Changes in global acetylation levels of lysine residues of histones H3 and H4 and H3 lysine 18 crotonylation following single and double *HDAC1/2* KO in BT67 cells ($n = 3$).
**h−m** Changes in the H4K5ac domains at 5′ regulatory regions of stem cell, cell-fate specific and the TGF-β pathway-related genes in *HDAC2* KO BTSCs relative to AAVS1 control cells. - represents molecular weight markers (50, 15, and 10). Source data are provided in the source data file.

HDAC2 and SMAD3 co-regulate the expression of genes associated with BTSC cell-fate, cell cycle progression and the TGF-β pathway-related genes.

We further investigated the functional relevance of the epigenetic and molecular overlaps between HDAC2 and SMAD3-SKI in BTSCs. Our results above show that chemical inhibition or genetic loss of HDAC2 decreased SMAD3 expression which correlated with decreased cell growth both in vitro and in vivo. We thus asked whether exogenous introduction of SMAD3 would rescue growth in HDAC2 deficient BTSCs. Overexpression of SMAD3 in *HDAC2* KO BTSCs rescued the expression of SOX2 compared to their respective *HDAC2*-/Neg+ control cells (Fig. 6a; Supplementary Fig. 18a, c, d). However, there was no apparent rescue of GFAP and BDNF expression in *HDAC2*-/SMAD3+ cells relative to *HDAC2*-/Neg+ control cells (Fig. 6a; Supplementary Fig. 18a, c, d). SMAD3 overexpression in HDAC2 proficient BTSCs (AAVS1/SMAD3+) led to a small increase in the protein levels of SOX2 but no change in BDNF and GFAP relative to the AAVS1 control BTSCs (Fig. 6a; Supplementary Fig. 18a). Taken together, these results indicate that SOX2 expression in BTSCs is, at least in part, regulated by SMAD3.

Furthermore, overexpression of SMAD3 significantly rescued the sphere forming frequency of *HDAC2* KO BTSCs relative to the *HDAC2*-/Neg+ control cells (Fig. 6b, Supplementary Fig. 18b). Further analysis of cell cycle changes revealed a significant increase in the percentage of *HDAC2*-/SMAD3+ BTSCs in S phase of the cell cycle and rescued the G2/M phase cell cycle arrest compared to *HDAC2*-/Neg+ cells (Fig. 6c; Supplementary Fig. 18e). These findings suggest that SMAD3 can partially rescue the self-renewal and cell cycle defects induced by loss of *HDAC2* and that this may, in part, be attributed to SMAD3-mediated rescue of SOX2 expression. Using co-IP assays with a SMAD3 antibody, we found a small increase in AAVS1/SMAD3+ while there was no pull down of HDAC2 in *HDAC2*-/Neg+ and *HDAC2*-/SMAD3+ cells (Fig. 6d), further supporting the importance of this association between HDAC2 and SMAD3 in the maintenance of BTSC properties.

Finally, we asked if SMAD3 overexpression-mediated rescue of cell cycle progression would also revert the tumorigenic potential of *HDAC2* KO BTSCs in vivo. AAVS1 control, *HDAC2*-/Neg+, AAVS1/SMAD3+ and *HDAC2*-/SMAD3+ BTSCs were orthotopically xenografted in SCID mice. We observed a significant reduction in the median survival of *HDAC2*-/SMAD3+ xenografted mice compared to *HDAC2*-/Neg+ xenografted mice (Fig. 6e). Mice xenografted with AAVS1/SMAD3+ BTSCs showed no change in survival compared to AAVS1 control mice. Collectively, these findings suggest that in *HDAC2* deficient BTSCs, SMAD3 overexpression can partially revert BTSC growth in vitro and in vivo. However, the functional impact of HDAC2 loss/inhibition is much broader than that associated with SMAD3 alone. Our above data suggest that this may be due to HDAC2 mediated impact on global histone acetylation and crotonylation patterns as well as epigenetic modulation of the TGF-β pathway-related genes.

## Specific inhibition of HDAC2 or SMAD3 impacts BTSC stemness

We next evaluated the importance of targeting SMAD3 in BTSCs by generating CRISPR-cas9 KOs. We found prominent decreases in the protein levels of the stem cell markers; SOX2 and OLIG2 and of SKI in *SMAD3* KOs relative to AAVS1 control BTSCs (Fig.7a; Supplementary Fig. 19a), confirming their regulation by SMAD3 as seen in the ChIP-seq data. A small reduction in the expression of HDAC2 protein was also observed with a concomitant increase in the protein levels of BDNF (Fig. 7a; Supplementary Fig. 19a). Further increases in the levels of H3K27ac and H4K5ac marks were observed, which could be attributed to a decrease in HDAC2 protein. This, in turn, resulted in decreased immunoprecipitation of HDAC2 protein with SMAD3 antibody and a concomitant reduction in its enrichment levels at 5′ end of *BDNF* gene in *SMAD3* depleted BTSCs compared to AAVS1 control (Fig. 7c; Supplementary Fig. 19c, d). Furthermore, *SMAD3* KO significantly reduced the self-renewal ability of BTSCs relative to the AAVS1 control cells (Fig. 7d). This was consistent with the in vivo findings where the mice orthotopically xenografted with *SMAD3* KO BTSCs showed improved overall survival relative to their respective AAVS1 control cells (Fig. 7e, Supplementary Fig. 19b). These findings confirm the role of SMAD3 in regulating the stem and tumorigenic properties of BTSCs through both HDAC2-dependent and -independent processes.

We next screened a recently available inhibitor, Santacruzamate A, with high potency to HDAC2, in BTSCs and HFAs to determine its therapeutic value. We found that the Santacruzamate A was slightly less potent than romidepsin but was sufficient to decrease cell viability at nanomolar doses in all the BTSC lines tested (Fig. 7f). Treatment of BTSCs with Santacruzamate A resulted in similar changes in the HDAC2-mediated histone modifications, associated gene expression and in the modulation in the TGF-β pathway-related genes as seen with romidepsin treatments or genetic knockout studies (Fig. 7g–i, Supplementary Fig. 20a–e). Taken together, these findings further confirm the functional relevance of HDAC2 in BTSCs.

## HDAC2 overexpression results in increased growth and self-renewal in normal NSCs

We next asked whether HDAC2 alone would be sufficient to induce cell functional changes in normal neural stem cells. We stably over-expressed HDAC2 in hiPSCs derived SOX2+/nestin+ NSCs (Fig. 8a). Interestingly, HDAC2 overexpression in NSCs resulted in concomitant upregulation of SMAD3 and SOX2 proteins, with a downregulation of BDNF protein levels (Fig. 8b; Supplementary Fig. 21c–e). Further evaluation of the effect of HDAC2 overexpression on the self-renewal ability of NSCs revealed a small increase in the number of neuro-spheres in HDAC2+ NSCs relative to negative control (Supplementary Fig. 21a). Most importantly, there was a significant increase in the total number of cells counted 5 days after seeding HDAC2+ NSCs relative to negative control NSCs (Fig. 8c). Further, EdU cell cycle analysis revealed a significant increase in the percentage of cells entering the S phase of cell cycle with a concomitant drop in the percentage of cells in G0/G1 cell cycle phase (Fig. 8d; Supplementary Fig. 21b). Collectively,

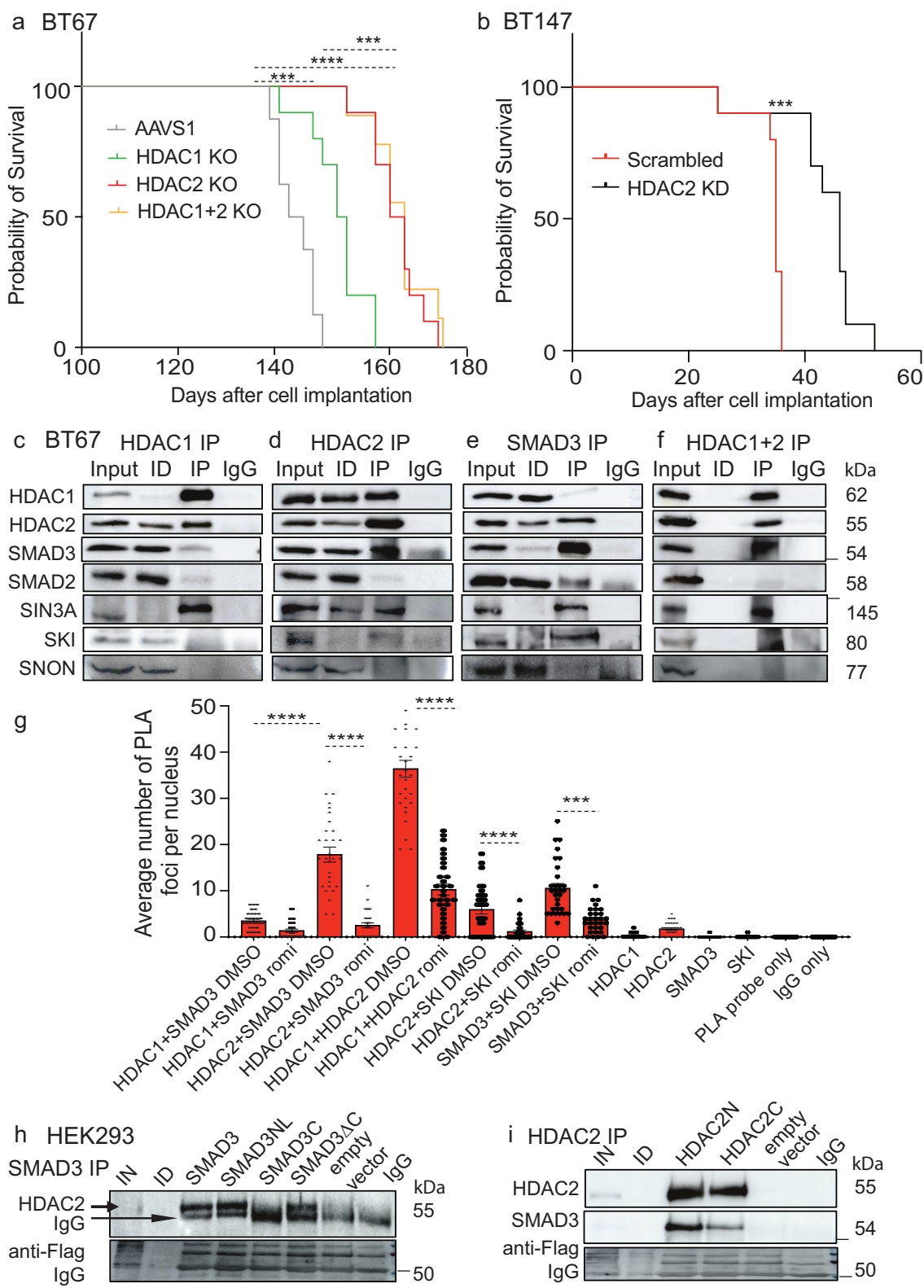

the data from HDAC2 gain-of-function in NSCs (Fig. 8d; Supplementary Fig. 21b) and loss-of-function in BTSCs (Fig. 3f; Supplementary Fig. 8f) suggest an important role of this essential epigenetic regulator in cell cycle progression. This effect could be due to HDAC2-mediated regulation of key stem cell (*SOX2* and *SMAD3*) and lineage-specific (*BDNF*) genes in both NSC and BTSC models. However, additional genetic and molecular aberrations such as mutations in EGFR, p53, PTEN or other

oncogenic driver events[2–4,30,32,45] may be needed along with HDAC2 dysregulation for complete cellular transformation of normal stem cells into GBM like tumors.

## Discussion

In this study, we show that HDAC2 displays context-specific and coordinated epigenetic activities, both independently and with other

**Fig. 4 | HDAC2 interacts with the TGF-β related proteins, SMAD3-SKI in BTSCs.**
**a** Kaplan–Meier survival curves for mice orthotopically xenografted with single and double *HDAC1/2* KO BT67 cells compared to AAVS1 control mice (Log-rank (Mantel–Cox) method, **$p < 0.01$, ****$p < 0.0001$, $n = 10$). **b** Kaplan–Meier survival curves for mice orthotopically xenografted with HDAC2 KD BT147 cells compared to mice xenografted with their respective scrambled control cells (Log-rank (Mantel–Cox) method, ****$p < 0.0002$, $n = 10$). **c–f** Co-immunoprecipitation and sequential co-IP assays in dual crosslinked BT67 cells. ($n = 3$). **g** In situ proximity ligation assays (PLA) confirm protein-protein interactions between HDAC2 and the SMAD3-SKI complex which was disrupted following 24 h treatment with romidepsin relative to DMSO control in BTSCs. Quantitative representation of the average number of PLA foci counted in 30 nuclei/reaction in BT67. Significance was determined using ANOVA (Tukey's test) at 95% confidence intervals, **$p < 0.01$, ***$p < 0.001$; Data are represented as mean values ± SEM; $n = 3$. HDAC1/2 protein-protein interaction was used as positive control and PLA probe only and IgG only were used as negative controls for the assay. **h, i** Co-IP assays in dual crosslinked HEK293T/17 cells expressing different SMAD3 (SMAD3, SMAD3NL, SMAD3C and SMAD3ΔC) and HDAC2(HDAC2N and HDAC2C) mutants or empty vector ($n = 3$ biologically independent transductions). Input: 1% input was used for all the co-IP assays. IP-immunoprecipitation, ID-immunodepleted samples, IgG- antibody controls for non-specific pull down. - represents molecular weight markers (50 and 150). Source data are provided in the source data file.

transcription regulators for the maintenance of stemness, growth, cell-fate related and, tumorigenic features of BTSCs. We show that HDAC2 regulates chromatin modifications that impact the expression of SMAD3 and SOX2 and is thus essential for BTSC self-renewal and growth. Furthermore, a protein-protein interaction between HDAC2 and the SMAD3-SKI proteins regulates the mechanisms which promote the cell cycle progression while inhibiting the expression of genes related to cell-fate specifications in BTSCs. Our data suggest that these pleiotropic roles of HDAC2, driven by specific chromatin modifications and modulation of the SMAD3-SKI arm of the TGF-β pathway, are critical for maintaining the stemness phenotype and growth of BTSCs (Fig. 8e).

The class I HDACs, particularly HDAC1/2, have been previously reported to be essential for regulating cell cycle progression and self-renewal of hESCs[22,46] and also display cell-type specific epigenetic regulations in glial versus neuronal cell lineages during different stages of neurodevelopment[24,47,48]. Here, we pinpoint the specific roles for HDAC1/2 in BTSCs and show that targeted inhibition of HDAC1/2 with romidepsin impacts BTSC cell viability, self-renewal and cell cycle progression, and transcriptional mechanisms associated with these cellular responses. These results parallel previously identified roles for HDAC1/2 in normal hESCs[22] and lead us to propose that BTSCs may regain or retain some of the epigenetic regulations of normal hESCs driven by either one or both HDAC1/2.

Recently, HDAC1 knockdown has been shown to impact the glioma stem cell phenotype, however the effects of HDAC1 knockdown were only observed in glioma stem cells with p53-wildtype status[28]. Here, our combined data with chemical inhibition, including the HDAC2-specific inhibitor Santacruzamate A, and genetic KO strategies show that while both HDAC1 and HDAC2 contribute to the growth characteristics of BTSCs, HDAC2 is the most relevant HDAC for growth and maintenance of stemness and tumor-initiating potential in BTSCs of different genetic backgrounds. Consistent with these observations, *HDAC2* KO resulted in a greater impact on global histone acetylation and crotonylation levels at the chromatin level. Specifically, the changes in the levels of H4K5ac and H3K18Cr were more strongly impacted by *HDAC2* KO. Broader domains of active histone modifications such as H3K4me3 have been shown to preferentially mark cell-fate specific genes[49], which could be the case for the broader domains of H4K5ac in regulation of stem-cell and fate-related programs in BTSCs. The link between genetic deletion of *HDAC1/2* and changes in the global H3K18Cr levels at the vicinity of transcriptional start sites of target genes including the stem cell genes, *Nanog* and *Pou5f1* has been previously reported by Kelly et al. in murine ESCs model[35].

There is limited evidence supporting specific epigenetic functions or cooperative molecular mechanisms of the most critical HDAC enzymes in cancer stem cell contexts. Here, we uncovered a protein-protein interaction between HDAC2 and SMAD3-SKI proteins, suggesting HDAC2-mediated epigenetic modulation of the transcriptional activity of SMAD3-SKI proteins in BTSCs. Interestingly, the expression of the master transcription factor, SMAD3 and the negative TGF-β regulator, SKI, were also impacted by the loss of HDAC2 in BTSCs. The TGF-β/SMAD pathway has been implicated in several different aspects of GBM pathogenesis, and treatment related drug-resistance[36,50–54]. However, the current approaches of targeting the critical components of the TGF-β pathway are not very conclusive[54], which may be partially attributed to epigenetic mechanisms, described here and previously[55], that influence the behavior of the TGF-β pathway.

Intriguingly, overexpression of SMAD3 in *HDAC2* KO BTSCs could not fully rescue the survival benefits of targeting HDAC2 in orthotopic xenograft BTSC models although genetic loss of *SMAD3* or specific pharmacological inhibition of HDAC2 were equally effective at altering the stem cell features of BTSCs. These findings suggest that HDAC2 may regulate additional molecular mechanisms, other than SMAD3-SKI, that are essential for BTSC growth and tumorigenesis. We were able to promote self-renewal and growth in normal NSCs by HDAC2 overexpression alone. Interestingly, we observed an increase in the protein abundance of the stem cell regulator, SOX2 and of the transcriptional regulator, SMAD3. These findings further confirm that the stem cell-specific aspects of BTSCs are regulated by HDAC2 promoting SMAD3, in a manner similar to normal NSCs.

Overall, our study shows epigenetic and molecular interactions between HDAC2 and SMAD3-SKI that are required for the maintenance of the key functional characteristics of BTSCs. We postulate that this and other such interactions may not be limited to GBM BTSCs but may also be relevant in other cancer stem cell contexts. Our findings suggest that the disruption of HDAC2-SMAD3-SKI axis with the use of HDAC2-specific brain penetrant small molecule inhibitors, has potential as a promising and global therapeutic strategy for targeting the drug-resistant and self-renewing BTSC population in GBM.

## Methods

This research complies with all relevant ethical regulations and all experimental procedures were conducted in accordance with the University of Calgary Ethics Review Board, Health Research Ethics Board of Alberta, and the Animal Care Committee of the University of Calgary.

### Cell culture

Patient-derived BTSC lines were cultured from a series of tumor specimens obtained, following written informed consent from patients, including identifying information such as sex and age (Table 1), during surgical resection and approval from the University of Calgary Ethics Review Board, Health Research Ethics Board of Alberta - Cancer Committee (HREBA) and Arnie Charbonneau Cancer Institute Research Ethics Board (REB HREBA-CC-160762) (Calgary, AB, Canada) as previously described[29–32,56]. All cell lines were cultured in stem cell enriched media supplemented with EGF (20 ng/ml; Peprotech), bFGF (20 ng/ml; R&D Systems Inc) and heparin sulfate (2 μg/ml; R&D Systems Inc) and were used within 10 passages after being thawed. Normal human brain tissues were obtained from 12- to 18-week-old fetuses from therapeutic abortions according to ethical guidelines established by the University of Calgary. Written parental consent was obtained from all the donors of tissues and the use of these human samples is approved by the institutional Review Board of the University of Calgary (REB14-1789). Tissue was collected from the health providers,

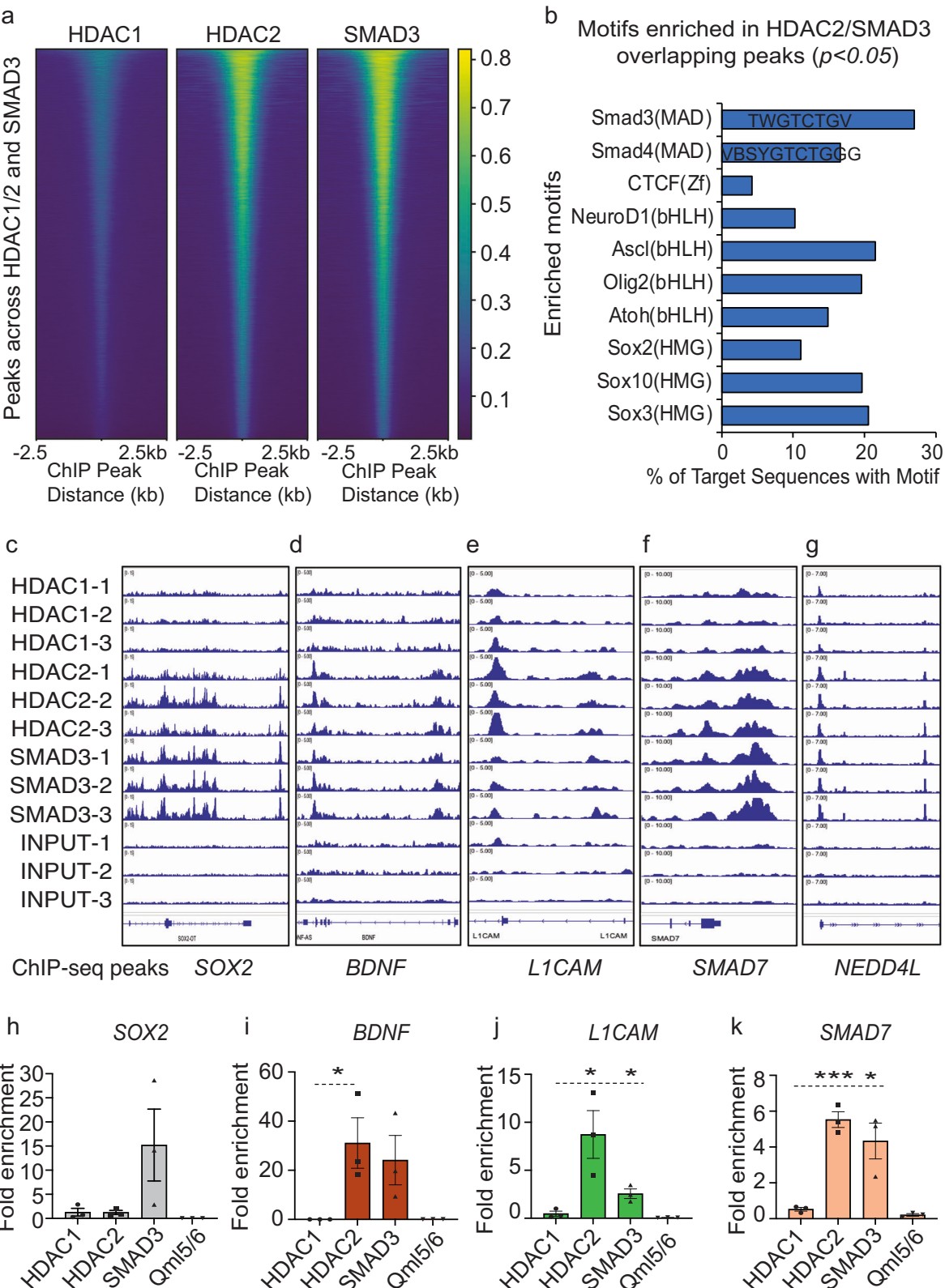

**Fig. 5 | HDAC2 and SMAD3 coregulate stemness and cell-fate programs and the TGF-β pathway-related genes in BTSCs. a** ChIP-sequencing evaluating the common target genes regulated by the HDAC1/2 and SMAD3 complex in BT67 cells. *n* = 3. **b** Motif enrichment analysis of HDAC2 and SMAD3 overlapping peaks. *p < 0.05 for all enriched motifs. **c−g** Enrichment levels of HDAC1/2 and SMAD3 within the gene regions of *SOX2, BDNF, L1CAM, SMAD7,* and NEDD4L.

**h−k** Validation of ChIP-seq data by ChIP-qPCR. Significance was determined using unpaired two-tailed *t*-test 95% confidence intervals, data are represented as fold enrichment mean values ± SEM; *p < 0.039, *p < 0.03 and p < 0.021, ***p < 0.00039 and *p < 0.019, n = 3). Fold enrichment of HDAC2 and SMAD3 within the heterochromatin region QML5/6 was used as negative control for ChIP-qPCR. Source data are provided in the source data file.

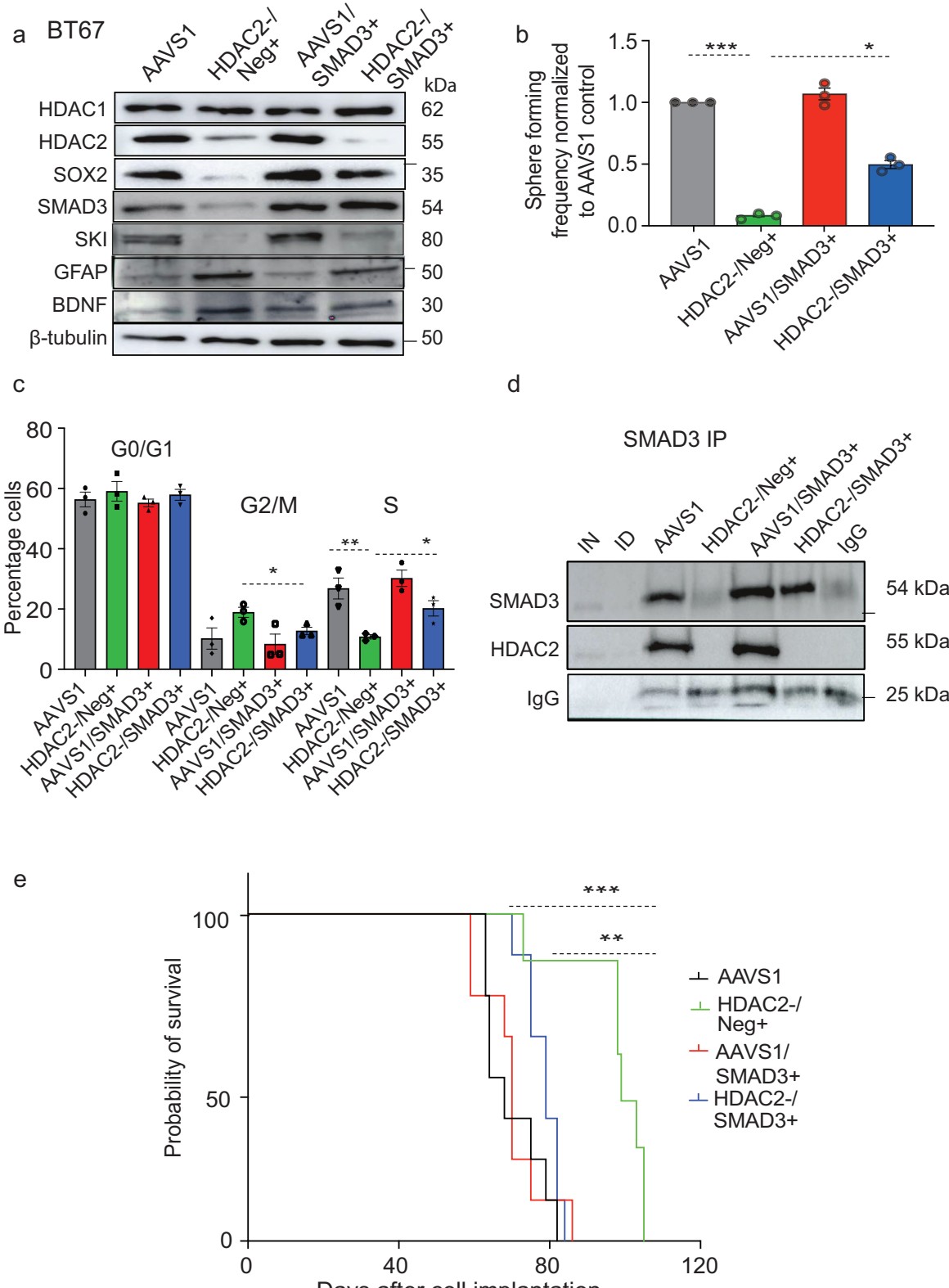

transported to the laboratory under sterile conditions and processed immediately for culture as previously described[57,58]. In brief, fetal brain tissues were minced and treated with DNase (Roche, Nutley, NL) followed by treatment with trypsin (Invitrogen, cat# T4049). The dissociated cells were then passed through a nylon mesh and Human fetal neural (HF-NSCs) stem cells were generated and cultured under stem cell enriched conditions similar to the BTSCs. Authenticated human

induced pluripotent stem cells (hiPSCs) were obtained from ATCC (ATCC-ACS-1020) and were cultured using similar materials and methods as for the human embryonic stem cells as previously described[59]. HEK293T/17 were obtained from ATCC (ATCC 293T/17) and the astrocytes derived from HF-NSCs and hiPSCs-NSCs were cultured in the Dulbecco's Modified Eagle Medium (DMEM) with 10% fetal bovine serum (FBS). All BTSC lines were authenticated using short

**Fig. 6 | HDAC2 and SMAD3 co-regulate self-renewal versus cell-fate specification potentials of BTSCs. a** Constitutive overexpression of SMAD3 in *HDAC2* knockout and AAVS1 control BT67 cells. Empty control vector was used as a negative control. Western blot analysis of protein levels of SOX2, GFAP, BDNF and SKI in HDAC2⁻/Neg⁺ vs. HDAC2⁻/SMAD3⁺ cells (*n* = 3). **b** Sphere forming frequency measured following SMAD3 overexpression in HDAC2 KO and AAVS1 control BT67 cells. Significance was determined using ANOVA (Tukey's test) at 95% confidence intervals, ***$p < 0.001$, *$p < 0.05$, Data represent mean ± upper and lower 95% confidence intervals, *n* = 3. **c** Analysis of cell cycle changes in AAVS1 control, HDAC2⁻/Neg⁺, HDAC2⁻/SMAD3⁺, AAVS1ᶠSMAD3⁺ BT67 cells. Significance was determined using ANOVA (Tukey's test) at 95% confidence intervals, *$p < 0.05$, **$p < 0.01$; data are represented as mean values ± SEM; *n* = 3. Gating strategies are provided in Supp. Fig. 22a. **d** Co-IP assays in dual crosslinked AAVS1 control, HDAC2⁻/Neg⁺, HDAC2⁻/SMAD3⁺, AAVS1/SMAD3⁺ BT67 cells (*n* = 3). **e** Kaplan–Meier survival curves for mice orthotopically xenografted with HDAC2⁻/SMAD3⁺ BT67 cells relative to the mice xenografted with HDAC2⁻/neg⁺ BT67 cells. (Log-rank (Mantel–Cox) method, **$p < 0.0026$, ***$p < 0.0002$, *n* = 8). · represents molecular weight markers (50, 37, and 25). Source data are provided in the source data file.

tandem repeat profiling and their profile was compared to the original parental tumors tissue (Calgary Laboratory Services and Department of Pathology and Laboratory Medicine, University of Calgary). Authentication and testing of all cell lines were performed as per American Association for Cancer Research recommendations. All cell lines used in this study were routinely tested for Mycoplasma using Vendor™ GeM Mycoplasma PCR-based detection kit (Sigma, MP0025) and they were confirmed to be mycoplasma free.

### Drug Screening and cell viability assays

Both pan- (Pracinostat cat# S1515) and Class I and II specific inhibitors (Romidepsin cat# S3020, Mocetinostat cat# S1122, 4SC202 cat# S7555, RGFP966 cat# S7229, WT161 cat# S8495 and Santacruzamate A cat# S7595) were obtained from Selleckchem. The pan HDAC inhibitor Panobinostat and the HDAC1 and 2 specific inhibitor romidepsin were obtained from Structural Genomic Consortium (SGC), University of Toronto, Ontario. BTSC lines and normal human astrocytes derived from HF-NSCs and NSCs were seeded in 96-well plates at a density of 1000–2500 cells/well (estimated based on growth rate of BTSCs and normal astrocytes). Twenty-four hours later, cells were treated with the respective drugs using a range of 0–3 μM or 0–1 nM (romidepsin) and the concentration of the vehicle was determined based on the highest concentration of each drug used for a period of 14 days.

For the HDAC1/2 CRISPR-cas9 knockout and shRNA knockdown cell viability assays, the cells were seeded 1000 cells/well in 96 well plates for 14 days. All cell viability assays were performed using the alamarBlue™ cell viability reagent (Thermo Fisher Scientific, cat# DAL1025), according to the manufacturer's protocol, for the desired incubation time of 6–8 h (hrs) (based on the growth rate of each cell line and to avoid signal saturation) and fluorescence was measured (excitation 540 nm and emission 590 nm). Result represent means from three biological replicates, each consisting of 3–6 technical replicates. Statistical analysis was conducted using nonlinear regression curve fit in GraphPad Prism 8.

### Limiting dilution assays

For assessing the effect of drug treatments or CRISPR-cas9 KO of target genes on the sphere-forming ability of BTSCs, limiting dilution assays (LDAs) were used as previously described[4]. Briefly, BTSCs from different experimental groups were plated in 96-well plated in a 10 × 6 grid with 10 columns of serial dilution from 512 cells to 1 cell per well (6 technical replicate wells per cell number). Plates were scored between 14–21 days by counting the number of wells that formed at least one sphere with a minimum diameter of 50 μm, per condition. The percentage sphere forming frequency was calculated using the Extreme Limiting Dilution Analysis Webtool[60]. Spheres were imaged using the IncuCyte ZOOM live-cell imaging instrument (Essen BioScience) and were acquired and exported using the IncuCyte ZOOM controller software (Essen BioScience, ver. 2016B, IncuCyteDRC package). Average sphere initiating frequency as well as upper and lower 95% confidence interval were plotted using GraphPad Prism 8. For each experimental condition, three biological replicates were performed.

### Neurosphere assays and cell counts

Briefly, the self-renewal ability of HDAC2⁺ NSCs and NSCs (empty vector) was assessed using previously described methods[59]. Cells were seeded in ultralow attachment 96 well plates at clonal density of (5–10 cells/μl) and total number of spheres/well were counted after 14 days for HDAC2⁺ NSCs relative to the empty control vector expressing NSCs. Spheres were collected individually from each well, dissociated, and total number of viable cells were counted using Trypan blue viability staining. For each experimental condition, 3 biological replicates were performed, and results were analyzed using GraphPad Prism 8.

### Annexin V cell survival and EdU cell cycle analysis

The cell viability in treated and untreated BTSC lines was assessed using Annexin V staining using PE Annexin V Apoptosis Detection Kit as per manufacturer's guidelines (BD Biosciences). Briefly, the BTSC cells were dissociated and $1.0 \times 10^5$ cells were resuspended in binding buffer and stained with Annexin V-Allophycocynin (APC) and 7-amino actinomycin D (7AAD) for 15 mins.

For cell cycle analysis, cells were dissociated and $2.5 \times 10^5$ cells/sample were exposed to 10 μM EdU for 1 h at 37 °C. Cells were fixed and stained using Click-iT™ Plus EdU Alexa Fluor™ 647 Flow Cytometry Assay kit (Thermo Fisher Scientific cat# C10634) as per manufacturer's guidelines. Cells were then counterstained with propidium iodide (PI) for DNA content using FxCycle™ PI/RNase Staining solution (Invitrogen cat# F10797). Flow cytometry analysis was performed using CytoFLEX LX (Beckman Coulter) at Flow cytometry core facility (University of Calgary) and results were analyzed using FACSDiva software version 6.1.3 (BD Biosciences).

### CRISPR-cas9 knockout procedures

CRISPR-cas9 technique was used to knockout HDAC1 and/or 2 or SMAD3 in BTSC lines using a previously described protocol[4]. Briefly, the BTSC lines were transduced using Cas9 virus, followed by blasticidin selection. The selected cells were then transduced using guide RNAs to target HDAC1 and/or 2 or SMAD3 followed by selection using puromycin. Two independent guide RNAs were used for HDAC1, 2, SMAD3 and AAVS1 was used as a cut control. For generating double knockouts, the puromycin-selected HDAC1 KO cells were subsequently transduced using HDAC2 guide RNAs. The knockout efficiency of each guide RNA was validated by western blot. The gRNA sequences used in this study are listed in Supplementary Table 1.

### Stable lentiviral knockdown and overexpression methods

Stable single and double knockdown of HDAC1 and/or 2 were performed using three independent shRNA sequences for each gene along with their respective scrambled shRNA controls (GeneCopoeia). The lentiviral particles were generated as previously described[4]. The shRNA sequences for targeting HDAC1 and HDAC2 were tagged with green- (GFP) and red-fluorescent proteins (RFP) as a marker of transduction efficiency. HDAC1 knockdown cells were selected using puromycin and hygromycin was used for selection of HDAC2 knockdown cells. HDAC1 and 2 double knockdown cells generated a yellow

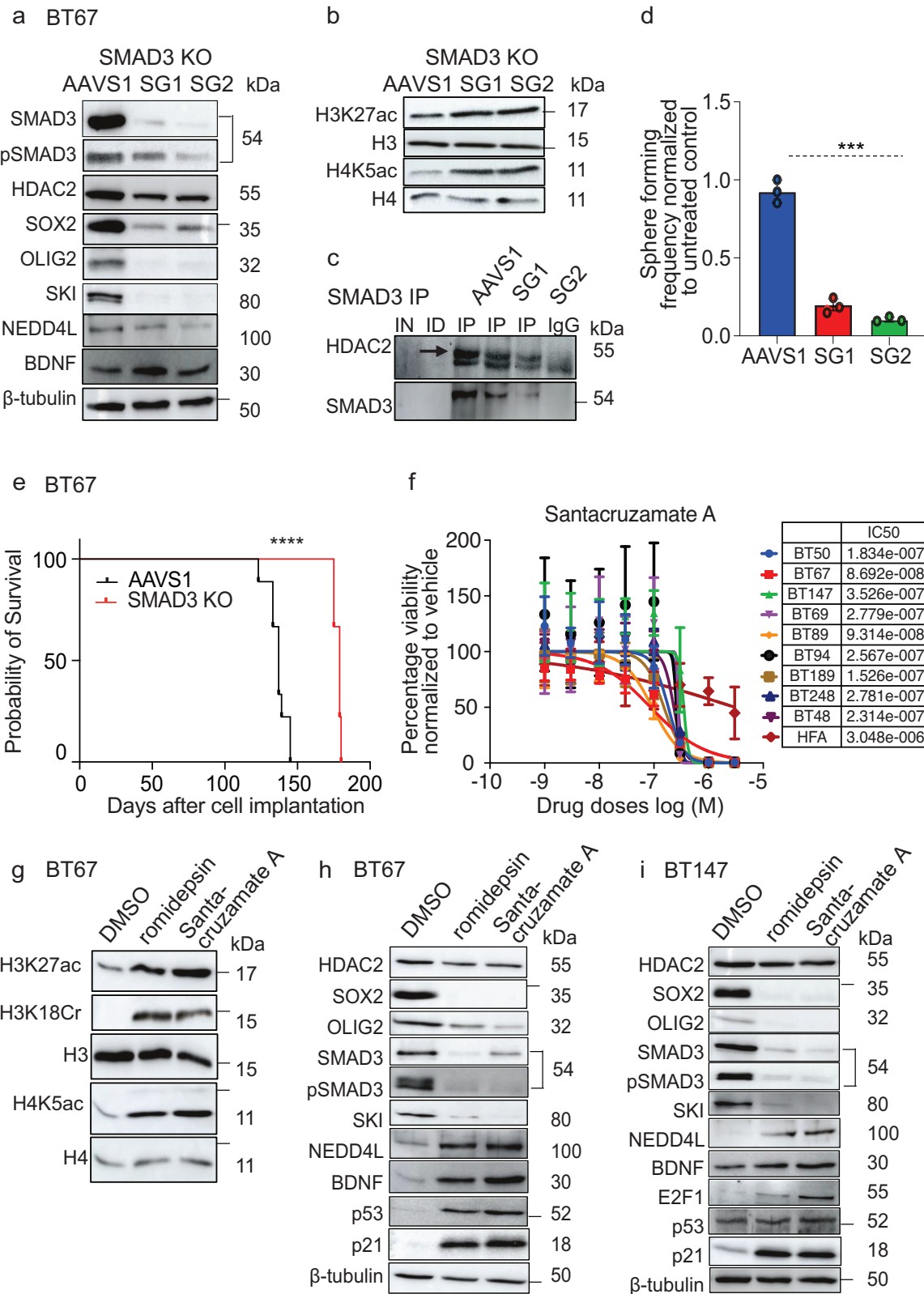

fluorescent signal and knockdown efficiency of each shRNA sequence was validated by western blot.

Stable overexpression of SMAD3 in HDAC2 KO cells was performed using LentiORF clone (Ex-L0088-Lv205-GS) along with GFP-expressing empty vector control clone (Ex-Neg-Lv205) from GeneCopoeia and lentiviral particles were generated as previously described[4]. GFP was used as a marker of transduction efficiency and SMAD3+ cells

were selected using puromycin. HDAC1 or 2 overexpression in double KO BT67 cells and HDAC2 overexpression in SOX2+/nestin+ NSCs was conducted using ready to use lentiviral particles (LPP-I0488-Lv207, LPP-Z8312_Lv207) from GeneCopoeia along with GFP-expressing empty control vector lentiviral particles (LPP-NEG-Lv207). Stable overexpression of custom designed mutant clones was performed in HEK293T/17 cells using mammalian pReceiver-M46 vector with C-flag

**Fig. 7 | Specific inhibition of HDAC2 or SMAD3 impacts BTSC stemness.**
**a** Western blot validation of CRISPR-cas9 mediated KO of *SMAD3* and its effect on
HDAC2, SOX2, OLIG2, SKI, NEDD4L, and BDNF protein levels in BTSCs relative to
AAVS1 control cells. **b** Changes in global H3K27ac and H4K5ac following *SMAD* KO
in BT67 cells ($n = 3$). **c** Co-IP assay in dual crosslinked AAVS1 control and *SMAD3* KO
BT67 cells ($n = 3$). **d** Sphere forming frequency measured following *SMAD3* KO in
BT67 cells relative to AAVS1 control cells. Significance was determined using
unpaired two-tailed *t*-test, ***$p < 0.001$; Data represent mean ± upper and lower 95%
confidence intervals, $n = 3$. **e** Kaplan–Meier survival curves for mice orthotopically
xenografted with *SMAD3* KO BT67 cells compared to AAVS1 control mice (Log-rank
(Mantel–Cox) method, ****$p < 0.0001$, $n = 10$). **f** Cell viability for 9 BTSC lines and

HFAs from three biological replicates following treatment with HDAC2 specific
inhibitor Santacruzamate A (100–1000 nM) Data represent mean values ± SD, $n = 3$.
**g** Acetylation levels of lysine residues of histones, H3 and H4, and crotonylation of
H3 at lysine 18 in BT67 cells following treatment with Santacruzamate A (100 nM)
and vehicle control for 72 h ($n = 3$). **h, i** Western blots showing changes in protein
levels of SOX2 and OLIG2, neuronal fate-specific marker; BDNF, the TGF-β pathway-
related proteins; total and phospho-SMAD3, SKI, and NEED4L and the cell cycle
proteins; p53, p21 (**h, i**) and E2F1 (**i**) following 72 h treatment with Santacruzamate A
(100 nM) and vehicle control ($n = 3$). - represents molecular weight markers (50, 37,
15, and 10). Source data are provided in the source data file.

tag and neomycin selection marker for SMAD3 (CS-U0365-M46-01-04)
and HDAC2 (CS-Z8312-M46-01-02) along with an empty control vector
(EX-Neg-M46). Stable overexpressing cells were selected using
hygromycin/neomycin. Overexpression of HDAC1/2 and SMAD3 was
validated by western blot.

## RNA-sequencing and bioinformatic analysis
The BTSC line, BT67, was treated with romidepsin and vehicle con-
trol for 72 h and pellets were collected from three biological repli-
cates. Total RNA was extracted using the RNeasy Kit (Qiagen, cat#
74104), quantified using Nanodrop, and sent for library preparation
and RNA-sequencing (Centre for Health Genomics and Informatics,
University of Calgary). Bioinformatic analysis was performed (Uni-
versity of Calgary) for determining differentially expressed genes
using Qiagen Ingenuity Pathway Analysis (IPA). Transcripts differ-
entially expressed at least 1.5-fold (up or downregulated) in romi-
depsin treated vs. vehicle treated BT67 cells and with a $p < 0.05$ were
considered significant. Gene set enrichment analysis was performed
using a preranked list of differentially expressed genes (FDR < 0.05)
between romidepsin and vehicle control. Significantly enriched
gene sets were visualized using cytoscape with enriched map and
autoannotate plug-ins.

## Real-time quantitative PCR
Total RNA was extracted and purified from treated and untreated cell
pellets using the RNeasy Kit (Qiagen, cat# 74104) and reverse tran-
scribed to cDNA using the qScript cDNA Synthesis Kit (Quantabio,
cat#95047). The cDNA was added to the reaction mixture containing
FastStart Essential DNA Green Master Mix (Roche, cat# 06402712001)
and 1.0 µmol/L forward and reverse primers. RT-qPCR was performed
in triplicate wells using the LightCycler 96 Instrument I (Roche). The
relative expression values were obtained using the LightCycler
96 software (Roche, ver. 1.1.10.1320) and normalized to Glycer-
aldehyde 3-phosphate dehydrogenase (GAPDH). All the PCR primers
used in the study are listed in Supplementary Table 2.

## Total histone extractions
Total histone extractions were performed using the acidic histone
extraction method as previously described[4]. Briefly, BTSC pellets were
collected from drug treatment and CRISPR-cas9 knockout experi-
ments, lysed and pellets containing nuclear content were resuspended
in cold 0.2 N $H_2SO_4$. Supernatants containing total histone extracts
were precipitated, washed, and re-suspended in 60 µl distilled $H_2O$
with 4X Laemmli buffer and 10 µL of 1 M Tris (pH 8), to adjust the pH.

## Western blot
Total histone samples and total protein samples extracted using RIPA
buffer were used for western blotting as previously described[59]. The
pellets for total protein extraction were collected from different
experimental treatments and. Blots were probed with the target and
controls antibodies overnight at 4 °C, following by washing and incu-
bation with horseradish peroxidase (HRP)-conjugated secondary
antibodies. Blots were imaged with SuperSignal and Amersham Imager

600 (General Electric). All the antibodies used against histone and non-
histones antigens are listed in Supplementary Table 4.

## Co-immunoprecipitation assays
BTSC CRISPR-cas9 KO lines or BT67 and BT147 treated with romi-
depsin and DMSO for 24, 48, and 72 h for time course study and for
48 h for the rest of the experiments were collected and were dual
crosslinking with dithiobis (succinimidyl propionate)) for 30 min and
with 1% formaldehyde for 10 min. Dual crosslinking was performed to
ensure the crosslinking of unmodified and monophosphorylated
forms of HDAC2[19]. Crosslinked pellets were snap frozen and stored at
−80 °C until further use. Cells were lysed in IP buffer (50 mM Tris-HCl,
pH 8.0. 150 mM NaCl, 1.0 mM EDTA, 0.5% Nonidet P-40) containing
phosphatase and protease inhibitors, and immunoprecipitations
reactions (IP) were performed as previously described[59]. Briefly, 500 µg
of total cell extract was incubated with 5 µg of different antibodies
overnight at 4 °C. The following day, 50 µL of protein G magnetic
dynabeads™ (Thermo Fisher Scientific cat# 10003D) were added for
3–4 h at 4 °C. Beads were then washed, proteins were eluted, and
immunoblotting was performed as described earlier[59]. For sequential-
IP the eluted proteins were diluted in IP buffer and incubated with
second antibody. An immunoprecipitation reaction was performed
with isotype-specific non-related IgG as a negative control to check the
nonspecific immunoprecipitation. An immunodepleted (ID) fractions
corresponding to each immunoprecipitation reaction was included to
test the immunoprecipitation efficiency. All the antibodies used are
listed in Supplementary Table 4.

## Polyubiquitination assay
BT67 cells were treated with romidepsin for 24 and 48 h along with
vehicle control and were processed to detect ubiquitinated-SMAD3
using polyubiquitination assay kit (Pierce™ Ubiquitin enrichment kit
Cat# 89899) as per the manufacturer's protocol. Briefly, lysates con-
taining 0.2 mg of total protein from romidepsin-treated or vehicle-
treated BT67 cells were incubated with 20 µl of polyubiquitin affinity
resin in spin column for overnight at 4 °C followed by centrifugation
and washing of the column containing ubiquitinated proteins. Ubi-
quitinated proteins were eluted, and immunoblotting was performed
as described earlier[59].

## Chromatin immunoprecipitation-sequencing and ChIP-qPCR
The formaldehyde crosslinked BT67 cells were used for H3K27ac and
H4K5ac Chromatin immunoprecipitation (ChIP)-sequencing. For
HDAC1/2 and SMAD3 specific (ChIP)-sequencing, dual crosslinked BT67
cells were used. Cell pellets were processed as previously described[59]
and sent for sequencing (Centre for Health Genomics and Informatics,
University of Calgary). Briefly, the cell pellets were suspended in cell lysis
buffer (5 mM PIPES (pHed with KOH to 8.0), 85 mM potassium chloride
(KCl), 0.5% NP-40) with freshly added phosphatase and protease inhi-
bitors. Nuclei were pelleted, resuspended in 1–2 mL Mnase digestion
buffer (10 mM Tris-HCl pH 7.5, 0.25 M sucrose, 75 mM sodium chloride
(NaCl)) plus phosphatase and protease inhibitors and were lysed by
adding 10% SDS (0.3% final concentration). Lysates were then sonicated

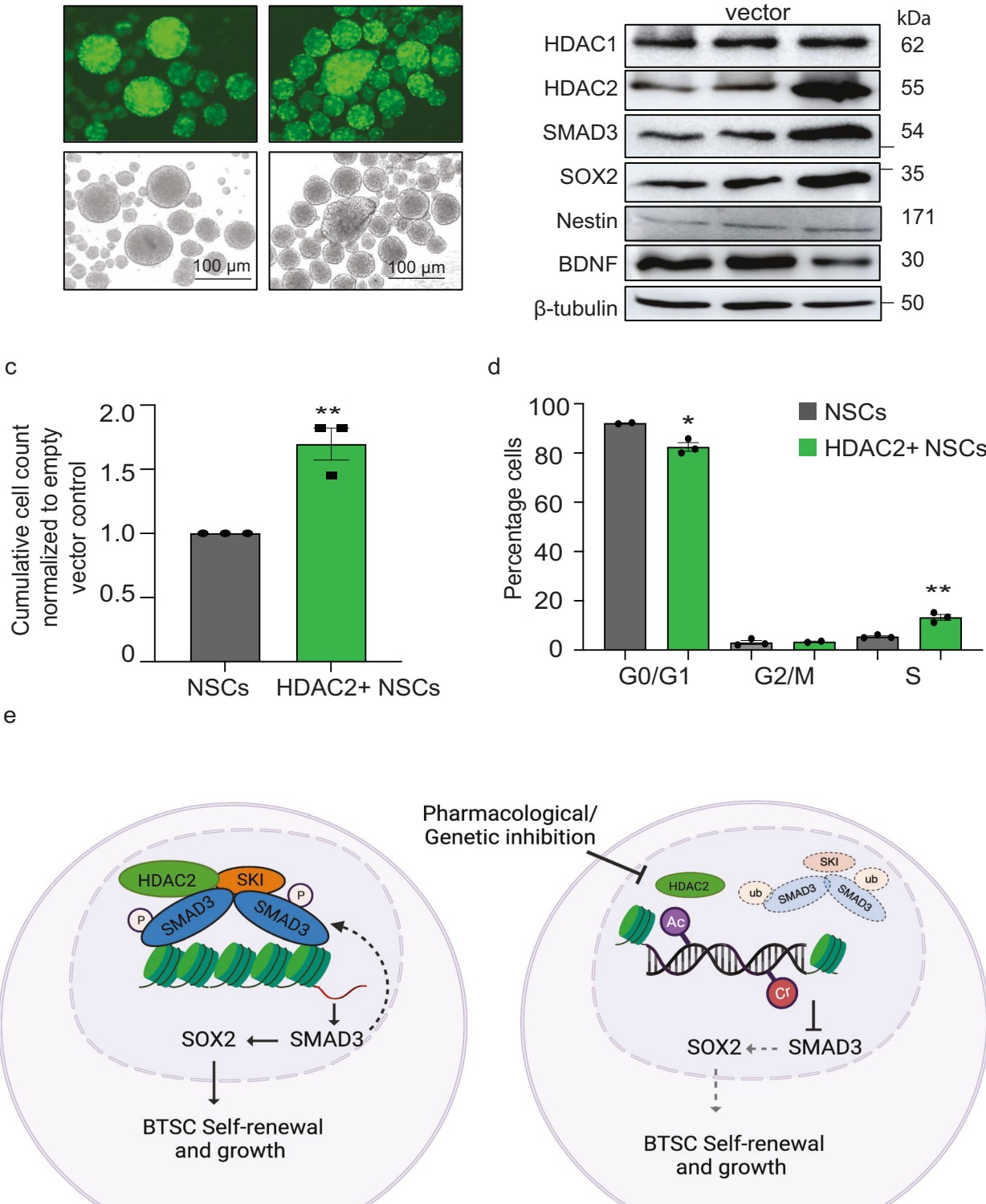

**Fig. 8 | HDAC2 Overexpression results in increased growth and self-renewal in normal NSCs. a** Overexpression (OE) of HDAC2 in hiPSCs derived SOX2$^+$/nestin$^+$ NSCs Scale bar: 100 μm. **b** Protein levels of stem cell and differentiation markers. **c** Cumulative cell counts 5 days after seeding cells at clonal density of (5−10 cells/μl) in neurosphere culture conditions. Significance was determined using unpaired two-tailed *t*-test, **$p$ < 0.0048; data are represented as mean values ± SEM; $n$ = 3. **d** An EdU analysis showing the changes in cell cycle progression in HDAC2$^+$ NSCs compared to empty vector control cells. Significance was determined using

unpaired two-tailed *t*-test, *$p$ < 0.023, **$p$ < 0.003; data are represented as mean values ± SEM; $n$ = 3. Gating strategies are provided in Supp. Fig. 22a. **e** Schematic model representing HDAC2-mediated epigenetic modulation of the transcriptional activity and expression of SMAD3-SKI proteins critical for the maintenance of BTSC stemness and tumorigenic potentials. - represents molecular weight markers (50 and 37). The model figure was generated using Biorender. Source data are provided in the source data file.

and tested for desired chromatin size between 200–1000 bp. Samples were diluted 1:5 in ChIP-RIPA buffer (10 mM Tris-HCl pH 8.0, 1% Triton-X 100, 0.1% Sodium dodecyl sulfate (SDS), 0.1% Sodium deoxycholate (SDC)) with freshly added phosphatase and protease inhibitors and DNA A260 (OD) were measured using a spectrophotometer. Two A260 of each cell lysate was used and incubated with 1–2 µg/A260 of different antibodies overnight at 4 °C. Antigen-antibody complex was pulled down using protein G magnetic dynabeads™ (Thermo Fisher Scientific cat# 10003D) were added for 3–4 h at 4 °C. The unbound chromatin was washed with low salt (0.1% SDS, 1% Triton-X-100, 2 mM EDTA, 20 mM Tris-HCl pH 8.1, 150 mM NaCl), high salt (0.1% SDS, 1% Triton-X-100, 2 mM EDTA, 20 mM Tris-HCl pH 8.1, 500 mM NaCl), lithium chloride (250 mM LiCl, 1% NP-40, 1% deoxycholate, 1 mM EDTA, 10 mM Tris-HCl pH 8.1) and then, Tris-EDTA (TE) (10 mM Tris-HCL pH 7.5, 1 mM EDTA) buffers. The bound fraction was eluted in 250 µl elution buffer (1% SDS, 100 mM NaHCO$_3$), reverse crosslinked and the DNA was extracted using a PCR purification Kit (QIAquick cat# 28106). RT-qPCR was performed on immunoprecipitation DNA using ChIP-PCR specific primers designed either to determine the changes in H3K27ac peaks near 5′ regulatory regions of genes of interest or within the regions where HDAC2 and SMAD3 peak were detected. All the PCR primers used in the study are listed in Supplementary Table 3.

### PCR cycling conditions

| Steps | Temperature | Time | Cycles |
|---|---|---|---|
| Initial denaturation | 98 °C | 30 s | 1 |
| Denaturation | 98 °C | 10 s | 25–35 cycles |
| Annealing | 55 °C | 20 s | |
| Extension | 72 °C | 10 min | |
| Final extension | 72 °C | X min | 1 |
| Hold | 4 °C | Forever | |

### Proximity ligation assays

In situ proximity ligation assays were performed to detect protein-protein interactions between the proteins of interest using (PLA) Duo-link® Kit (cat# DUO92002, Sigma-Aldrich) containing red detection reagents and Mouse minus and Rabbit plus probes as per the manufacturer's protocol. Briefly, cells were cultured adherently on microscope glass slides with hydrophobic wells coated with poly-L-ornithine (Sigma-Aldrich, cat# P4957) and laminin (Sigma-Aldrich, cat# L2020) using similar conditions and media as used for 3D BTSCs cultures[30]. The adherent monolayer of cells was treated with romidepsin or vehicle control for 24 h and was fixed for 30 min at RT in 3.7% formaldehyde, followed by permeabilization in 0.25% Triton X-100 solution for 10 min and blocking (Duolink kit) for 30 min at 37 °C. Primary antibodies were diluted in the antibody diluent provided (Duolink kit) and were applied to respective wells for 1 hr at RT. The wells were washed using 2× wash buffer (Duolink kit) and PLA probes were added according to the manufacturer's protocol for 1 h at 37 °C. Following incubation with probes, the wells were washed with wash buffer and the ligation was carried out at 37 °C for 30 min. The wells were washed 2× in wash buffer A (Duolink kit). Amplification step was carried out in dark for 100 min at 37 °C as per the manufacturer's protocol and then wells were washed 2× in wash buffer B and 1× in 0.01× wash buffer B. Slides were air dried, cover slipped using the provided mounting medium containing DAPI and imaged on a high-capacity Olympus VS120 slide scanner at 40× magnification. A minimum of 30 nuclei were processed for PLA foci quantification using the Fiji Image analysis tool (ImageJ2 v2.14.0) with consistent noise level settings for all images (University of Calgary). All the antibodies used are listed in Supplementary Table 4.

### Intracranial xenografts

All animal procedures were performed as previously described[43,61,62] with the animal ethics protocol (AC21-0162) approved by the Animal Care Committee of the University of Calgary and operating in accordance with the Guide to the Care and Use of Experimental Animals published by the Canadian Council on Animal Care and the Guide for the Care and Use of Laboratory Animals issued by NIH (Bethesda, MD). 6–8 weeks old CB-17 female SCID mice were housed in Biohazard barrier levels 2 facility at a temperature 25+/−2 °C, 45–55% humidity, and a light cycle of 6am on, 8 pm off. All animals used were female, and as such no sex-based analysis was performed. Mice were anesthetized and 100,000 BT67 and (BT67 AAVS1, HDAC1, HDAC2 or HDAC1 + 2 KOs; BT67 HDAC2 KD or scrambled control; BT67 AAVS1, AAVS1/SMAD3+, HDAC2-/Neg+ or HDAC2-/SMAD3+; BT67 AAVS1 or SMAD3 KO) and 50,000 BT147 (BT147 HDAC2 KD or scrambled control) cells were dissociated and were stereotactically injected into the right cerebral hemisphere of SCID mice (10 mice/group) at the depth of 3.5 mm using a 10 L Hamilton syringe. Endpoint was reached once animals showed any signs of neurological symptoms such as ataxia, >15% weight loss, hunching, kyphosis, paresis and lethargy, poor oral intake and domed heads. A lethal dose of ketamine/xylazine (Ketamine 300–360 mg/kg + xylazine 30–40 mg/kg) followed by cervical dislocation (as approved under our institutional animal certification) was used to euthanize animals. Survival data was analyzed using Log Rank (Mantel–Cox) in GraphPad Prism and plotted as Kaplan–Meier survival curves.

### Statistical analysis

All experiments were performed using $n = 3$ biological and $n = 3$–6 technical replicates. Descriptive statistical analyses were performed using Microsoft Excel (Version 16) and GraphPad Prism (Version 8) to determine significant differences using unpaired two-tailed Student $t$-test or ANOVA followed by Dunnett or Tukey's test for multiple comparisons and data are reported as mean ± SEM. Values of $p < 0.05$ were considered statistically significant. For orthotopic xenograft experiments, median survival was measured according to the Kaplan–Meier method, with a Log Rank Mantle–Cox test to compute the statistical significance for 10 mice per treatment group.

### ChIP-sequencing data processing and bioinformatic analysis

ChIP-seq reads were first trimmed for adapter sequences and low-quality score bases using Trimmomatic[63]. The resulting reads were mapped to the human reference genome (hg38) using BWA-MEM[64] in paired-end mode at default parameters. Only reads that had a unique alignment (mapping quality >20) were retained and PCR duplicates were marked using Picard tools (https://broadinstitute.github.io/picard/). Peaks were called using MACS2 software suite[65]. A "reference peak set" was obtained by merging ChIP-seq peaks from each samples using bed tools merge with parameters: -sorted -d −125 (https://bedtools.readthedocs.io/). Peaks were associated with the nearest TSS of genes (+/−20 kb) using the annotate Peaks command from HOMER software suite[66]. In addition, peak enrichments were calculated as Fragments Per Kilobase of transcript per Million mapped reads (FPKM). Genome browser tracks were created with the HOMER make UCSC file command and bed Graph To Big Wig utility from UCSC. Tracks were normalized so that each value represents the read count per base pair per 10 million reads. UCSC Genome Browser (http://genome.ucsc.edu/)[67] was implemented for track visualization. Motif enrichment was performed using the findMotifsGenome command.

### Reporting summary

Further information on research design is available in the Nature Portfolio Reporting Summary linked to this article.

## Data availability

The bulk RNA-seq data for BT67 romidepsin treated versus DMSO treated cells, H3K27ac ChIP-seq data for BT67 romidepsin treated

versus DMSO treated cells, H4K5ac data for HDAC2 KO and AAVS1 control BT67 cells and HDAC1/2 and SMAD3 ChIP-seq data for BT67 cells are deposited in NCBI's Gene Expression Omnibus (GEO) repository accession number: GSE214721 and GSE214926. The remaining data are available within the Article, Supplementary Information or Source Data file provided with this manuscript. Source data are provided with this paper.

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

## Acknowledgements

This study was supported by a grant from the Canadian Institutes for Health Research (153246 to H.A.L. and S.W.). Weiss was also supported by a Stand Up to Cancer Canada Cancer Stem Cell Dream Team grant (SU2CAACR-DT-19-15) with funding provided by the Government of Canada through Genome Canada and the Canadian Institutes of Health Research. Stand Up To Cancer Canada is a Canadian Registered Charity (Reg. # 80550 6730 RR0001). Research Funding is administered by the American Association for Cancer Research International – Canada, the Scientific Partner of SU2C Canada. We thank Dr. Wee Yong's lab at the University of Calgary for providing human fetal brain tissue samples. We thank Yiping Liu Facility staff at The Flow Cytometry core facility (University of Calgary) for Annexin V and EdU cell cycle flow cytometric analysis. We thank the Centre for Health Genomics and Informatics (University of Calgary) for performing the RNA- and ChIP-sequencing and Dr. Paul Gordon, Bioinformatics Manager at Centre for Health Genomics and Informatics for analyzing RNA-sequencing data. We thank François Lefebvre, Bioinformatics Manager and Alain Pacis, Bioinformatics Specialist at The Canadian Center for Computational Genomics (C3G) for performing bioinformatic analysis for all the ChIP-sequencing data used in the manuscript. The Canadian Center for Computational Genomics (C3G) is a Genomics Technology Platform (GTP) supported by the Canadian Government through Genome Canada. The authors thank Drs Marco Gallo and Shirin Bonni from University of Calgary for editorial and scientific input.

## Author contributions

R.B., Conceptualization, study design, data collection, analysis and interpretation, validation, manuscript-writing original draft, review and editing. X.H., In vivo experiments and analyses, IHC, manuscript-review. R.H., Establishing and maintaining BTSCs cultures, data collection, methodology. O.C., Data collection, validation, methodology, manuscript-review. D.A.B., designing guide RNA sequences for CRISPR-cas9 studies, manuscript-review. H.A.L., Conceptualization, supervision, funding acquisition, methodology, validation, project administration, manuscript-editing and review. S.W., Conceptualization, resources, supervision, funding acquisition, validation, methodology, project administration, editing, and final approval of manuscript.

## Competing interests

The authors declare no competing interests.
