## [Peer Review File · Nature Communications]

Epigenetic and molecular coordination between HDAC2 and SMAD3-SKI regulates essential brain tumour stem cell characteristics.Reviewers' Comments:

Reviewer #1:

Remarks to the Author:

In this manuscript by Bahia et al., examine the role of histone deacetylases in GBM brain tumor stem cells (BTSC) using pharmacological (romidepsin) as well as genetic manipulation and multiple experimental strategies (histone crotonylation, ChIP assays for changes in histone marks, RNA-seq analysis and orthotopic tumor growth). The authors conclude that HDAC2 is the most relevant HDAC for BTSC growth and self-renewal. In addition, the authors have identified that HDAC2 interacts with SMAD3 and SKI proteins (members of TGF β pathway) and that this interaction is necessary for the tumorigenic potential of BTSCs. The authors conclude that inhibition of HDAC2 activity and disruption of mechanisms regulated by HDAC2-SMAD3-SKI are promising therapeutic avenues for targeting GBM BTSCs. Despite some of the detailed analyses conducted there was a disconnect between the central message and experimental strategies. Furthermore, authors have to be mindful that phenotypes obtained with pharmacological inhibition vs genetic knockout experiments can be very different and can be challenging to compare or draw any conclusions.

There are some issues that authors should address:

1. The central conclusion in the abstract is that HDAC2 is the most relevant HDAC for growth and self-renewal of BTSCs. Further experiments are required to solidify that statement. Initial experiments are conducted with romidepsin, which is not selective for HDAC2 and inhibits both HDAC1 and 2 at similar potency. CRISPR-mediated knockout experiments do not really address requirement of HDAC1 over HDAC2.
2. In Figure 3e-f and Supplemental Figure 6e-f knockout of both HDAC1 and HDAC2 does not result in greater decrease in viability (i.e. no additive effect) of BTSCs. Can authors provide an explanation for this result? The *in vitro* result is also observed *in vivo*, wherein HDAC1/2 double knockout does not result in prolonged survival relative to HDAC2 knock out alone. However, both HDAC1 and HDAC2 KO individually provide some benefit. The authors do not attempt to provide an explanation or speculate for why this might be the case.
3. Is the % reduction in viability significant when comparing HDAC2 KO to HDAC1 KO cells (figure 3d-f) - might be helpful to plot everything on the same figure panel/graph (HDAC1 KO vs HDAC2 KO vs double KO).
4. Can over-expression of HDAC1 or HDAC2 in the double knockout cells rescue the observed reduction in cell viability?
5. Are the levels of HDAC3, another HDAC isoform that can regulate histone acetylation, changing or compensating for loss of HDAC1/2?
6. There are differences in histone crotonylation observed after knockout of HDAC1 vs HDAC2. Authors should discuss the implications of this result further.
7. Is HDAC2 more important than HDAC1 under distinct genetic backgrounds? As authors have pointed out HDAC1 has been shown to play an important role in p53 WT cells. The two cell lines utilized here - BT67 -p53Wt and BT147 - P53mut also have other genetic aberrations. At least the way they have been plotted here don't seem to have dramatic differences in cell viability between HDAC1 vs HDAC2 KO.
8. Authors should perform the sphere forming assay in HDAC1 KO cells and compare this to HDAC2 KO, especially if they claim that HDAC2 is essential for the self-renewal of BTSCs similar to Figure 1d.
9. Only stem cell marker examined was SOX2. Authors should include other stem cell markers as well.
10. When the authors conclude that HDAC2 KO induces differentiation of BTSCs, it is not clear what cell type these cells are presumed to be differentiating into.
11. What is the putative mechanism that results in reduced cell viability after HDAC2 KO? How is this different from what is occurring in HDAC1 KO cells?
12. Are HDAC2 and HDAC1 part of distinct complexes with BTSCs? Have authors analyzed other remodeling factors that immunoprecipitated with SMAD3?
13. In the discussion, the authors should comment on romidepsin clinical trial in GBM which failed to meet its primary outcome and include possible discrepancies that can be observed between pharmacological inhibition of HDACs (e.g. lack of specificity, off-target effects) vs functional studies

using genetic knockouts. The use of romidepsin minimizes the findings in the latter half of the paper as the drug inhibits both HDAC1 and HDAC2 potently.

14. In the discussion, the authors should make an attempt to try and discuss their findings in the context of other results demonstrating that HDAC1 was also shown to be critical for BTSC growth and speculate on the observed differences across the two studies. It would be important to determine if specific genetic backgrounds influence the importance of HDAC2 over HDAC1 in BTSCs.

15. Individual values from repeats should be plotted for all bar graphs.

16. Provide the sex of the mice used for intracranial surgeries

Reviewer #2:

Remarks to the Author:

Summary

This study is essentially important in terms of cancer stem cell biology and molecular therapeutics. HDAC2-SMAD3 signaling may provide a new angle to understand the etiology of GBM. Also, HDAC2 is likely to be an attractive target for GBM treatment, in particular considering that small molecule HDAC inhibitors are currently available in clinical trials. On a broad perspective, HDAC2 regulation of SMAD3 expression and function could act as a fundamental mechanism that influences transcription and stemness in different cancer types. The authors have conducted extensive molecular and cellular experiments and drawn an elegant pathway involved in GBM stem cell function. A major limitation of this study is lack of mechanistic insights into the relation between HDAC2 and SMAD3.

Major comments:

1. HDAC2 is a known cofactor of SMAD3, but how they associate with each other remains unknown. Thus, it is important to map HDAC2-binding domain(s) in the SMAD3 protein. Also, HDAC1 has limited capability to bind SMAD3. It will be interesting to examine whether HDAC2 is required for the recruitment of HDAC1 to SMAD3.

2. It is unclear how HDAC2 and SMAD3 co-regulate gene expression in GBM BTSCs. First, are SMAD3-responsive motifs (such as SBE and CAGA) enriched within the HDAC2/SMAD3 overlapping sites (Supp Fig.11a)? Second, would SMAD3 depletion affect the DNA-binding activity of HDAC2? Third, would SMAD3 depletion affect the expression of HDAC2 target genes as well as their associated histone modifications (such as H3K27ac and H4K5ac)? Fourth, HDAC2/SMAD3 co-occupied genes are both up- and down-regulated upon romidepsin treatment. How to interpret these seemingly opposite effects? Lastly, as a control, HDAC1 ChIP-seq should be conducted to determine whether HDAC1/SMAD3 overlaps exist. Moreover, comparison of HDAC1 and HDAC2 ChIP-seq datasets could yield an additional line of evidence supporting that HDAC2 has more diverse roles than HDAC1 in BTSCs.

3. No evidence supports a direct link between HDAC2-regulated histone modifications and the fate of the TGF- β pathway genes. The authors should conduct H3K27ac and H4K5ac ChIP-qPCR on SOX2, SMAD3, SKI and BDNF in response to HDAC1 and HDAC2 KO (versus the AAVS1 control).

4. It is unclear whether romidepsin repression of SMAD3 predominantly relies on gene silencing or protein turnover. It will be important to determine whether the E3 ligase NEDD4L plays a key role in SMAD3 downregulation upon romidepsin treatment (Page 8; Lines 172-178).

5. It is unclear how romidepsin destabilizes HDAC2-SMAD3 interaction. Although this is likely because of SMAD3 downregulation, HDAC2 IP upon romidepsin treatment at different time points would be a critical experiment to clarify whether SMAD3 is dissociated from HDAC2 before the protein disappears. Also, it is unclear whether and how HDAC enzymatic activity contributes to HDAC2-SMAD3 interaction. Since p300/CBP/KAT5-mediated lysine acetylation plays a key role in SMAD3 function, it is important to assess whether SMAD3 becomes acetylated after the addition of romidepsin.

6. Technically, it is unclear how the authors conducted RNA-seq differential expression analysis – $\text{Log}_2(\text{fold change}) > 1.0$ or 1.5 ? How many genes are significantly up- or down-regulated (in Fig.1f)? Pathway and/or gene-set enrichment analyses are also missing.

7. Santacruzamate A (a selective HDAC2 inhibitor, PMID: 32839551) should be tested for verification of the romidepsin data.

8. The results presented here do not fully support the working model (Fig.6e). For instance, no experimental evidence suggests SMAD4 as part of the HDAC2-SMAD3 complex. Moreover, it remains to be determined whether romidepsin influences SMAD3 phosphorylation.

Minor comments:

1. Page 1; Line 2 – Title: cells → cell.
2. Page 3; Lines 55-56 - aberrant methylation or acetylation histone patterns → aberrant histone methylation or acetylation patterns.
3. Page 5; Line 110 – HFAs → human fetal astrocytes (HFAs).
4. Page 7; Lines 151-152 – Supp Fig. 2a → 4a.
5. Page 7; Line 152 – patterns → levels.
6. Page 7; Lines 161-163 – In addition to western blot (Fig. 2a and Supp Fig. 4c), the authors should validate RNA-seq data by RT-qPCR.
7. Fig. 4d – Compared to other IP experiments (Figs. 4a-c), why was no ID signal detected in HDAC1+2 IP?
8. The authors should conduct SKI IP to confirm its interaction with HDAC1/2 and SMAD3.
9. Supp Figs. 9a-b and 10a-b – Romidepsin treatment did not affect HDAC1 and HDAC2 protein expression in BTSC lines (Fig. 2a and Supp Fig.4c). But, why was the efficiency of HDAC1/2 IP under romidepsin treatment much lower than the vehicle control? This inconsistency might lead to an incorrect conclusion that romidepsin destabilizes the interaction between HDAC1/2 and SMAD3.
10. Supp Figs. 9d and 10d – To fully validate the IP data, PLA experiments should be conducted in both vehicle and romidepsin-treated BTSCs.
11. Fig. 4e – The cell line name was missing.
12. In ChIP-seq experiments, did HDAC2 and SMAD3 co-occupy the SOX2 locus? Since SOX2 is a key target of SMAD3, this finding should be validated by HDAC2/SMAD3 ChIP-qPCR and recruited to the main figure.
13. Page 14; Lines 328-329 – Compared to HDAC2-/Neg+, the median survival seemed to be reduced in the HDAC2-/SMAD3+ mice (Fig. 5n).
14. Would SMAD3 depletion affect the viability of BTSCs and the median survival of xenografted mice?
15. NSCs express HDAC2 at a level similar to that seen in BTSCs (Fig. 1a). In this regard, what is the rationale for using NSCs (but not HDAC2-negative/low astrocytes) in HDAC2 overexpression experiments (Figs. 6a-d)?

Reviewer #3:

Remarks to the Author:

Bahia et al present solid observations about the key role of HDAC2 in regulating brain tumour stem cells characteristics by performing in vitro and in vivo experiments as well as elucidating partially its mechanism of action. They propose that HDAC2 cooperates with SMAD3-SKI to maintain the tumorigenic potential in brain tumour stem cells. Inhibition of genetic manipulation of HDAC1/2 shows that HDAC2 has a more relevant role in regulating cell viability, self-renewal and differentiation by modulating chromatin accessibility. This work is of relevance to the field in order to understand why targeting the TGF-beta pathway in glioblastoma has not resulted in a better outcome for patients, this article indicates that epigenetic changes modulated by HDAC2 could influence TGF-beta pathway. Nonetheless, several important aspects of the manuscript need attention.

Major points

- Some of the western blots show clear results, others are not so clear; it will make it less speculative to show quantification of at least 3 western blots for all the immunoblots panels in the article (at least for Fig 2a-b, Fig 3a-c, Fig 5k, Fig 6b, Sup. Fig 4c,d,f, Sup. Fig 6a-c, Sup. Fig 12a,
- The authors validate by WB some of the differentially expressed genes found in the RNASeq analysis, such as SOX2, BDNF and GFAP. It would be interesting to check other stem cell fate regulators such as Olig1 and 2.

- The authors claim that the modulation in the components of the TGF- β pathway was specific to the inhibition of HDAC1/2 with romidepsin and was not observed with any other HDAC inhibitors; however in Supp Fig 4f they only check Smad3 and Smad7; they should also check P-Smad3, NEDD4L and SKI.
- The authors claim that the modulation in the components of the TGF- β pathway is mediated by HDAC2; however in Fig 3a and Supp Fig 6a they don't check Smad7; P-Smad3, and NEDD4L which were initially also shown to be modulated by HDAC1/2 inhibitors, and it would be important to check that those effects are not unspecific of the inhibitors.
- In order to show the importance of Smad3 loss upon HDAC2 KO, the authors rescue Smad3 levels by overexpression, showing that this rescues SOX2 levels and reduces GFAP and BDNF levels. It would be good to check here if Smad3 overexpression can also rescue the reduction of SKI observed upon HDAC2 KO shown in Fig3b-c.
- The authors could also check GFAP and BDNF expression in B147 after Smad3 overexpression (Supp Fig 12a) to complement their findings in BT67.
- Smad3 overexpression partially rescues the effects induced by loss of HDAC2. Is it because it does not rescue SKI? If so, would SKI overexpression together with Smad3 completely rescue from the effects induced by loss of HDAC2 in cell viability, cell cycle, self-renewal and differentiation?

Minor points

- Line 152: Should be Supp Fig 4a
- Line 158-159: please add references that show that those genes are involved in neurogenesis, synaptogenesis and neural stem cell fates (it would be useful for people who are new to the field)
- Line 326: I believe AAVS1/SMAD3+ is missing from this sentence
- Line 404 mentions several references that should talk about TGF-beta effects on GBM BTSCs, however some of them do not refer to this topic.
- Line 408: there is no reference 65 in the reference list. Please double check the references cited
- General comment for figure legends: please add the time of treatment for each assay (cell viability, limiting dilution assay, Western blot, etc) it will be easier for the reader than having to go back and forth to the method section.
- Fig 2 legend says phospho-p38 while the figure shows p38 and there is no phospho-p38 antibody listed in the data set file.
- Fig3j: in the figure legend is mentioned **** while the graph shows only 3. They should match
- Fig 5 legend caption: SKI levels are not checked; therefore, one can speculate that SKI is involved, but cannot assume. I think it would be better to remove SKI from the figure legend title. OR better, as mentioned in major points check SKI levels upon SMAD3 OE. And overexpress SKI together with S3 to see if overexpression of D3 and SKI together completely reverse HDAC2 KO effects in panel k, l and m

REVIEWER COMMENTS

Reviewer #1, expertise in BTSCs/GBM and HDAC inhibitors (Remarks to the Author):

In this manuscript by Bahia et al., examine the role of histone deacetylases in GBM brain tumor stem cells (BTSC) using pharmacological (romidepsin) as well as genetic manipulation and multiple experimental strategies (histone crotonylation, ChIP assays for changes in histone marks, RNA-seq analysis and orthotopic tumor growth). The authors conclude that HDAC2 is the most relevant HDAC for BTSC growth and self-renewal. In addition, the authors have identified that HDAC2 interacts with SMAD3 and SKI proteins (members of TGFb pathway) and that this interaction is necessary for the tumorigenic potential of BTSCs. The authors conclude that inhibition of HDAC2 activity and disruption of mechanisms regulated by HDAC2-SMAD3-SKI are promising therapeutic avenues for targeting GBM BTSCs. Despite some of the detailed analyses conducted there was a disconnect between the central message and experimental strategies. Furthermore, authors have to be mindful that phenotypes obtained with pharmacological inhibition vs genetic knockout experiments can be very different and can be challenging to compare or draw any conclusions.

We thank the reviewer for these valuable insights, and we have addressed the suggestions with experimentation and provided clarifications as outlined below:

There are some issues that authors should address:

1. The central conclusion in the abstract is that HDAC2 is the most relevant HDAC for growth and self-renewal of BTSCs. Further experiments are required to solidify that statement. Initial experiments are conducted with romidepsin, which is not selective for HDAC2 and inhibits both HDAC1 and 2 at similar potency. CRISPR-mediated knockout experiments do not really address requirement of HDAC1 over HDAC2.

We agree with the reviewer that romidepsin does not distinguish for selective inhibition of HDAC1 vs HDAC2. To further support the relevance of HDAC2 specific regulatory mechanisms in BTSCs, we performed following experiments:

- 1) Using HDAC1, HDAC2 and SMAD3 specific ChIP-seq, we observed increased binding of HDAC2 than HDAC1 in BTSCs genome with greater overlap between HDAC2 and SMAD3 binding. These findings further confirm the relevance of the epigenetic and molecular coordination between HDAC2 and SMAD3 in regulating the transcriptional programs in BTSCs. We have incorporated the new data in Fig 5 and Supp Fig 17 of the revised manuscript.
- 2) We next performed H4K5ac specific histone ChIP-seq using AAVS1 and HDAC2 KO BT67 cells. Our data revealed an overall change in the distribution of H4K5ac mark at the 5' end and genomic regions of genes regulated by HDAC2 (Please see Fig 3h-m, Supp Fig 10, 11). As suggested by the reviewer 2 in comment #3, we used H3K27ac and H4K5ac ChIP-qPCR to validate the changes observed in these histone modifications at the SOX2, SMAD3, SKI and BDNF gene loci, using CRISPR-cas9 KO BTSCs, particularly. The new data are presented in the Supp Fig 10, 11 of the revised manuscript.

3) Lastly, we validated the functional relevance of HDAC2 compared to HDAC1 by screening, the recently available, specific inhibitor Santacruzamate A in BTSCs (as suggested by reviewer# 2 in comment #7). The HDAC2-mediated specific effects on BTSC viability, histone acetylation levels and on gene expression including the modulation of TGF- β pathway related genes were similar to the results seen with romidepsin. These findings confirm that HDAC2 inhibition is sufficient for these mechanistic and phenotypic changes. The new data have been incorporated in the Fig 7e-i and Supp Fig 20 and we hope that these additional data help address reviewer's comment.

2. In Figure 3e-f and Supplemental Figure 6e-f knockout of both HDAC1 and HDAC2 does not result in greater decrease in viability (i.e. no additive effect) of BTSCs. Can authors provide an explanation for this result? The *in vitro* result is also observed *in vivo*, wherein HDAC1/2 double knockout does not result in prolonged survival relative to HDAC2 knock out alone. However, both HDAC1 and HDAC2KO individually provide some benefit. The authors do not attempt to provide an explanation or speculate for why this might be the case.

HDAC1 and 2 have a high degree of structural homology and have been shown to form homo- or hetero-dimers that allow them to function, in cooperation or separately, as part of multiprotein co-repressor complexes (Delcuva et al, 2012). Our *in vitro* and *in vivo* data lead us to propose that, in addition to their individual roles in BTSCs, loss of one or the other also impacts their cooperative functions. Our data thus suggest that HDAC2 single and double KOs results in more unique and additive effects including ~20% greater loss of cell viability compared to KO of HDAC1 alone. As per reviewer's suggestion in the next comment (#3), we have clarified these observations by plotting the data in one graph and performed statistical analysis accordingly. The results show a significant drop in both cell viability and sphere forming ability of BTSCs following single and double KO of HDAC2 relative to the HDAC1 KO BTSCs. Please see the revised data in Fig 3d, e and Supp Fig 8d, e.

3. Is the % reduction in viability significant when comparing HDAC2 KO to HDAC1 KO cells (figure 3d-f) - might be helpful to plot everything on the same figure panel/graph (HDAC1 KO vs HDAC2 KO vs double KO).

We thank the reviewer for bringing this to our attention. As per the suggestion, we have plotted the cell viability data for HDAC1 vs. HDAC2 single and double KOs in the same graph. As shown in this graph, HDAC2 single and HDAC1/2 double KOs have ~20% greater loss in cell viability compared to HDAC1 KO BTSCs. (Please see revised Fig 3d and Supp Fig 8d). The % reduction was significant when compared HDAC2 single or double KO BTSCs (for BT67 HDAC1+2-SG2 vs HDAC1-SG2 and for BT147 HDAC2-SG1 vs HDAC1-SG1) to HDAC1 KO. Furthermore, the impact on sphere forming abilities was significantly higher in HDAC2 KO BTSCs compared to HDAC1 KOs, which suggest that HDAC2 seems to influence the stem cell function of BTSCs. The new LDA data are included in the revised Fig 3e and Supp Fig 8e as per comment #8.

4. Can over-expression of HDAC1 or HDAC2 in the double knockout cells rescue the observed reduction in cell viability?

This is an interesting point and to address this question, we overexpressed HDAC1 or 2 in double KO BT 67 cells. The results are presented below

Unfortunately, one caveat of stable CRISPR-cas9 knockout system is that the gRNAs are constantly being generated and can target the exogenously introduced gene sequences. This makes it difficult to achieve the expression levels of HDAC1 or 2 comparable to the original protein levels and to evaluate an evident rescue of BTSC phenotype in a CRISPR-cas9 KO background (as depicted above).

Thus as an alternate strategy, we performed a rescue experiment by overexpressing SMAD3 which we identified as one of the critical target genes and coordinating molecular partner of HDAC2 in BTSCs. Overexpression of SMAD3 partially rescued the functional deficits induced by loss of HDAC2 (Please see Fig 6, Supp Fig 18).

5. Are the levels of HDAC3, another HDAC isoform that can regulate histone acetylation, changing or compensating for loss of HDAC1/2?

We tested the HDAC3 specific inhibitor, RGFP966, which was effective in reducing viability at higher nanomolar doses, in all of the BTSC lines tested. However, the specific effects that were observed with HDAC1/2 specific inhibitors; romidepsin and Santacruzamate A, particularly on histone modifications such as H4K5ac and H3K18Cr and on TGF- β pathway related genes were not observed in the samples treated with RGFP966 and are shown in the Supp Fig 4e, f and the new data in Supp Fig 19a). Previous studies have suggested a greater structural homology of HDAC3 with HDAC1 (64%) than with HDAC2 (Luo and Li, 2020), which help explain these findings. Furthermore, HDAC1/2 have been described as forming multimeric co-repressor complexes such as mSin3 and NuRD whereas HDAC3 forms a different co-repressor complex (Methot et al, 2008; Delcuva et al, 2012). These studies suggest that there may be increased possibility of compensation with HDAC1 loss than with HDAC2.

6. There are differences in histone crotonylation observed after knockout of HDAC1 vs HDAC2. Authors should discuss the implications of this result further.

We agree that this interesting observation will require further investigations outside the scope of this manuscript. Based on the current observation, we speculate that like the previous study by Kelly et al, 2018, increased levels of H3K18Cr following HDAC2 inhibition may overlap with H3K18ac and indicate active gene transcription. However, the role of this histone modification in normal vs. disease context is not very clear to date. A statement reflecting this point has been added in the discussion section of the revised manuscript.

7. Is HDAC2 more important than HDAC1 under distinct genetic backgrounds? As authors have pointed out HDAC1 has been shown to play an important role in p53 WT cells. The two cell lines utilized here - BT67 -p53Wt and BT147 – P53mut also have other genetic aberrations. At least the way they have been plotted here don't seem to have dramatic differences in cell viability between HDAC1 vs HDAC2 KO.

We agree with the reviewer that additional genetic alterations may play a role in the observed discrepancies between these two studies, particularly, related to the p53-dependent role of HDAC1 in Glioma stem cells. Our data from multiple cell lines with different p53 status suggest that HDAC2 is critical in all BTSC lines tested for cell viability, irrespective of their p53 status (Table 1). In addition, treatment with the HDAC2 specific inhibitor, Santacruzamate A, increases the protein levels of E2F1 in p53mut BT147 cells, suggesting that an induction of p53-independent mechanism may be affecting cell viability in p53-mutant lines. As per the revised cell viability graph added for comment #3, HDAC2 single and HDAC1/2 double KOs have ~20% greater loss in cell viability compared to HDAC1 KO BTSCs. In addition, the impact on sphere forming ability was significantly higher in HDAC2 KO BTSCs compared to HDAC1 KOs. This was associated with a reduction in the protein levels of the stem cell markers, SOX2 and OLIG2. These findings suggest that HDAC2 has high relevance for maintenance of the key stem cell features of BTSCs in addition to their survival. The revised and new data are shown in Fig 3d, e and Supp Fig 8d, e as per comment #8; Fig 7h, i of revised manuscript.

8. Authors should perform the sphere forming assay in HDAC1 KO cells and compare this to HDAC2 KO, especially if they claim that HDAC2 is essential for the self-renewal of BTSCs similar to Figure 1d.

We thank the reviewer for this comment and have performed the limiting dilution assays (LDAs) as a measure of the self-renewal ability of HDAC1 vs HDAC2 KO BTSCs. We observed a dramatic reduction on sphere forming ability of HDAC2 single and HDAC1/2 double KOs vs HDAC1 KO, highlighting the relevance of HDAC2 for the self-renewal properties of BTSCs. (Please see the revised Fig 3e and Supp Fig 8e).

9. Only stem cell marker examined was SOX2. Authors should include other stem cell markers as well.

We examined the expression levels of an additional stem cell marker, OLIG2 and found a decrease in the protein levels of OLIG2 following pharmacological (using

Santacruzamate A) and genetic inhibition of HDAC2. The new western blot data have been added in the revised Fig 3a-c, Supp Fig 8a-c, Fig 7h, i; Supp Fig 20a).

10. When the authors conclude that HDAC2 KO induces differentiation of BTSCs, it is not clear what cell type these cells are presumed to be differentiating into.

We agree with the reviewer that differential gene expression data are not conclusive of terminal differentiation into neuronal lineages but rather a measure of the expression of markers associated with differentiated cell types. Here, based on the RNA-seq data, we see the expression of genes associated with neuronal differentiation and synapse formation such as BDNF (a known target of HDAC2; Guan et al, 2009), STX3, NRXN1 and NEFH1 go up. Further gene set enrichment analysis revealed an enrichment of gene sets associated with neuronal and synaptic functions which were upregulated in response to romidepsin treatment. Changes in the mRNA and protein abundance of BDNF and STX3 genes, in romidepsin treated and HDAC2 KO cells, confirm that loss of HDAC2 promotes the expression of genes associated with neuronal functions. We have therefore, revised the original statement in the manuscript to better reflect these observations in the result section. Please see the new GSEA data in Supp Fig5, revised Fig 3a-c and Supp Fig 8a-c.

11. What is the putative mechanism that results in reduced cell viability after HDAC2 KO? How is this different from what is occurring in HDAC1 KO cells?

Our data suggest that genetic or pharmacological inhibition of HDAC2 target many cellular and molecular processes in BTSCs. Loss of HDAC2 reduces self-renewal and promotes the expression of genes associated with cell-fate specification programs in BTSCs. It also induces changes in the BTSC cell cycle progression and these cells are arrested at G2/M phase of cell cycles, presumably, leading to loss of cell viability. These cellular effects are a consequence of HDAC2-mediated changes in specific histone modifications, associated gene expressions and disruption of its coordination with the SMAD3/SKI proteins following specific inhibition of HDAC2. Consistent with the previous findings by Lo Cascio et al, 2021, the effect of inhibition of HDAC1 seems to occur via p53-dependent (in WT BTSCs) manner as suggested from the new data added in the Fig 7h, i of the revised manuscript.

12. Are HDAC2 and HDAC1 part of distinct complexes with BTSCs? Have authors analyzed other remodeling factors that immunoprecipitated with SMAD3?

Our data from co-immunoprecipitation and proximity ligation assays show an interaction between HDAC1 and 2 in BTSCs. These data suggest that HDAC1 and 2 function both in cooperation as well as separately as part of multimeric co-repressor complexes. Since the modulation in the TGF- β pathway related genes was specific to HDAC2, we investigated this relationship in depth. However, we cannot rule out the possibility of additional unidentified remodelling factors, cooperating with SMAD3. Investigating this premise would requires comprehensive follow-up studies of SMAD3 specific interactome in BTSCs.

13. In the discussion, the authors should comment on romidepsin clinical trial in GBM which failed to meet its primary outcome and include possible discrepancies that can be observed between pharmacological inhibition of HDACs (e.g. lack of specificity, off-target effects) vs functional studies using genetic knockouts. The use of romidepsin minimizes the findings in the latter half of the paper as the drug inhibits both HDAC1 and HDAC2 potently.

We agree with the possible discrepancies between using pharmacological inhibition vs. genetic knockout studies. On the suggestion of reviewer #2, here we utilized one such recently available HDAC2 specific inhibitor, Santacruzamate A, and were able to reproduce HDAC2 specific unique effects observed with romidepsin or genetic knockout studies. While there are, to our knowledge, no such specific inhibitors yet available for HDAC1 and it remains to be seen whether Santacruzamate A could have preclinical and clinical relevance, these findings help support the relevance of HDAC2 in GBM pathogenesis, which were readily discernable with using romidepsin. (Please see the new data added in the Fig. 7e-i). The genetic studies performed here help advance the understanding of which class/ classes and the relevance of specific HDACs to the pathological implications in different diseases. Ideally, such studies will help push forward the design of more specific HDAC inhibitors, with minimal side effects, by the medicinal chemistry field and lead to the future generation of clinically relevant inhibitors.

14. In the discussion, the authors should make an attempt to try and discuss their findings in the context of other results demonstrating that HDAC1 was also shown to be critical for BTSC growth and speculate on the observed differences across the two studies. It would be important to determine if specific genetic backgrounds influence the importance of HDAC2 over HDAC1 in BTSCs.

We thank the reviewer for this point; we have added survival data showing that HDAC1 knockdown in p53-mutant BT147 cells did not improve overall survival in orthotopic xenografts (Please see Fig 13g). These data are consistent with the previous study on HDAC1's relevance in p53 wild-type backgrounds. Based on our data, we believe that HDAC2-SMAD3/SKI axis may be critical for HDAC2 specific effects in BTSCs, irrespective of p53-status. We have included additional statements to address this comment in the result and discussion sections.

15. Individual values from repeats should be plotted for all bar graphs.

As per the suggestion, we have revised the bar graphs or have added additional supplementary figures showing individual values from the repeats.

16. Provide the sex of the mice used for intracranial surgeries

Female mice were used for all the *in vivo* experiments. This information has been added in the manuscript.

Reviewer #2, expertise in epigenetics and HDAC inhibitors (Remarks to the Author):

Summary

This study is essentially important in terms of cancer stem cell biology and molecular therapeutics. HDAC2-SMAD3 signaling may provide a new angle to understand the etiology of GBM. Also, HDAC2 is likely to be an attractive target for GBM treatment, in particular considering that small molecule HDAC inhibitors are currently available in clinical trials. On a broad perspective, HDAC2 regulation of SMAD3 expression and function could act as a fundamental mechanism that influences transcription and stemness in different cancer types. The authors have conducted extensive molecular and cellular experiments and drawn an elegant pathway involved in GBM stem cell function. A major limitation of this study is lack of mechanistic insights into the relation between HDAC2 and SMAD3.

Major comments:

1. HDAC2 is a known cofactor of SMAD3, but how they associate with each other remains unknown. Thus, it is important to map HDAC2-binding domain(s) in the SMAD3 protein. Also, HDAC1 has limited capability to bind SMAD3. It will be interesting to examine whether HDAC2 is required for the recruitment of HDAC1 to SMAD3.

We thank the reviewer for this comment and agree that this is indeed a critical aspect to help enhance the study findings. To address this key question, we designed mutant ORF clones with flag-tag for SMAD3 and HDAC2. We used HEK293 cells to overexpress mutant clones and performed Co-IP assays to map HDAC2-binding domains in the SMAD3 protein. Our data show that the MH1 domain of SMAD3 seems to be essential for binding to the HDAC association domain of HDAC2 protein and these new findings have been incorporated in the Fig 4h-l and Supp Fig 14h-j. Furthermore, using the sequential-IP assay, where the protein complex was immunoprecipitated with HDAC1-, followed by HDAC2-antibody, shows immunoprecipitation of HDAC2-SMAD3/SKI proteins (Fig 4f). The findings above indicate that HDAC1 may also be part of this protein complex. More comprehensive proteomic analyses will help further validating whether HDAC1 and 2 function independently or as part of same multimeric complex in BTSCs.

2. It is unclear how HDAC2 and SMAD3 co-regulate gene expression in GBM BTSCs. First, are SMAD3-responsive motifs (such as SBE and CAGA) enriched within the HDAC2/SMAD3 overlapping sites (Supp Fig.11a)? Second, would SMAD3 depletion affect the DNA-binding activity of HDAC2? Third, would SMAD3 depletion affect the expression of HDAC2 target genes as well as their associated histone modifications (such as H3K27ac and H4K5ac)? Fourth, HDAC2/SMAD3 co-occupied genes are both up- and down-regulated upon romidepsin treatment. How to interpret these seemingly opposite effects? Lastly, as a control, HDAC1 ChIP-seq should be conducted to determine whether HDAC1/SMAD3 overlaps exist. Moreover, comparison of HDAC1 and HDAC2 ChIP-seq datasets could yield an additional line of evidence supporting that HDAC2 has more diverse roles than HDAC1 in BTSCs.

To address these points, we performed the following experiments:

- 1) We performed HDAC1, 2 and SMAD3 ChIP-seq to evaluate their genome wide distribution and overlapping sites. ChIP-seq data show greater overlap between HDAC2 and SMAD3 binding peaks than for HDAC1. HDAC2 and SMAD3 overlapping sequences are enriched with SMAD3, NEUROD1 and SOX family related motifs. The new ChIP-seq data have been added in the revised Fig 5 and Supp Fig 17.
- 2) We performed CRISPR-cas9 KO of SMAD3 in BTSCs and found a reduction in HDAC2 immunoprecipitation with SMAD3 antibody in KO samples relative to AAVS1 control (Please see Fig 7c). There was also reduced binding of HDAC2 at the gene loci of SOX2 and BDNF, further confirming the functional relevance of their association (Supp Fig 19c, d).
- 3) Our new data shows that depletion of SMAD3 reduced the protein levels of SOX2, OLIG2 and SKI genes. We also observed a small decrease in HDAC2 protein with a comparable increase in the BDNF levels (Please see revised Fig 7a, Supp Fig 19a). There was also an associated increase in the levels of H3K27ac and H4K5ac histone marks which could be due to a change in HDAC2 protein in SMAD3 depleted BTSCs relative to AAVS1 control (Fig 7b).
- 4) Our H4K5ac ChIP-seq data shows the broader distribution of this active histone mark within genomic regions of differentiation markers such as BDNF, STX3 and NEUROD1/2 with a reduction in H4K5ac enrichment at 5' regulatory regions of stem cell genes, SOX2 and OLIG1/2. Hence, similar to the function of H3K4me3 breadth (shown in previous study by Benayoun et al, 2014), we propose that the breadth of the H4K5ac mark within the genomic regions of gene associated with differentiation markers and the peak height at the 5' regulatory regions of stem cell genes may indicate important and opposing roles in gene expression associated with stemness versus differentiation. Furthermore, SMAD3 shows increased binding at the SOX2 locus whereas HDAC2 has increased binding at L1CAM gene regions. These data suggest that they have both cooperative and independent roles in BTSCs, which may contribute to the opposite effects on gene transcriptions in BTSCs. The new H4K4ac ChIP-seq data have been added in the revised Fig 3h-m and Fig 5c, h, e, j.
- 5) Our new ChIP-seq data, which have been added to the manuscript, suggest that HDAC1 does not have a significant overlap with SMAD3 in BTSC genome and further validate the importance of HDAC2 over the HDAC1 in BTSCs. (Please see revised Fig 5).

3. No evidence supports a direct link between HDAC2-regulated histone modifications and the fate of the TGF- β pathway genes. The authors should conduct H3K27ac and H4K5ac ChIP-qPCR on SOX2, SMAD3, SKI and BDNF in response to HDAC1 and HDAC2 KO (versus the AAVS1 control).

We thank the reviewer for this important suggestion. We performed H4K5ac ChIP-seq using HDAC2 KO BT67 relative to AAVS1 control cells and determined that HDAC2 KO cells showed a reduction in H4K5ac enrichment at the 5' regulatory regions of SOX2, SMAD3 and SKI relative to AAVS1 control cells. In contrast, we observed a broader distribution of H4K5ac at 5' regulatory end and within gene regions of BDNF, STX3,

NEFH, NEUROD1/2 and L1CAM genes in HDAC2 KO cells. We further performed H3K27ac and H4K5ac ChIP-qPCR to validate the changes in the enrichment levels of these histone marks at the regulatory regions of SOX2, SMAD3, SKI and BDNF genes. These findings indicate that HDAC2 regulated genomic distribution of H3K27ac and H4K5ac may play a role in stemness versus cell-fate specification abilities of BTSCs. The new H4K5ac CHIP-seq and H3K27ac and H4K5ac ChIP-qPCR data have been incorporated in the revised Fig 3h-m, Supp Fig 10 and 11).

4. It is unclear whether romidepsin repression of SMAD3 predominantly relies on gene silencing or protein turnover. It will be important to determine whether the E3 ligase NEDD4L plays a key role in SMAD3 downregulation upon romidepsin treatment (Page 8; Lines 172-178).

We appreciate the reviewers' point. Since there is no, commercially available, NEDD4L specific E3 ligase inhibitor, we have addressed the comments in the following manner:

- 1) We report a concomitant increase in the levels of NEDD4L protein with a gradual decrease in the levels of total and phospho-SMAD3 following a time course treatment with romidepsin. These new findings have been presented in the Supp Fig 14f.
- 2) Next, we assessed the protein levels of ubiquitinated SMAD3 using a polyubiquitinated assay kit and found an increase in ubiquitinated SMAD3 (Supp Fig 5f). Lastly, the ChIP-seq and ChIP-qPCR data reveal cooccurrence of HDAC2-SMAD3 at 5' end of NEDD4L genomic region. Please see the new ChIP-seq data in the revised Fig 5g and Supp Fig 17d.

These findings suggest that NEDD4L, in part, plays a role in downregulation of SMAD3 in BTSCs. (Please see Supp Fig 13b, c). Furthermore, inhibition of HDAC2 decreased mRNA expression and the enrichment levels of H3K27ac and H4K5ac at the 5' end of SMAD3 gene as shown in Fig 1f, Fig 3l, Supp Fig 7a, c, Supp Fig 10g, k and Supp Fig b, f, suggesting its repression at gene transcription level.

5. It is unclear how romidepsin destabilizes HDAC2-SMAD3 interaction. Although this is likely because of SMAD3 downregulation, HDAC2 IP upon romidepsin treatment at different time points would be a critical experiment to clarify whether SMAD3 is dissociated from HDAC2 before the protein disappears. Also, it is unclear whether and how HDAC enzymatic activity contributes to HDAC2-SMAD3 interaction. Since p300/CBP/KAT5-mediated lysine acetylation plays a key role in SMAD3 function, it is important to assess whether SMAD3 becomes acetylated after the addition of romidepsin.

As per the reviewer's suggestion:

- 1) We treated BT67 cells with romidepsin and collected samples at different time points (24, 48, 72 and 96 hours post treatment) for Co-IP and western blot to assess whether changes in the protein abundance of SMAD3 play a role in decreasing its interaction with HDAC2. Our data suggest that the interaction between HDAC2 and SMAD3 is disrupted before the levels of SMAD3 protein decrease. The decrease in

the levels of SMAD3 correlates with a concomitant increase in NEDD4L protein, which may, in part, be responsible for the degradation of SMAD3 protein. Please see the new data added in the Supp Fig 14f, g.

- 2) Since the available HDAC enzymatic activity assay kits are non-specific, we overexpressed mutant clones of SMAD3 and HDAC2 in HEK293 cells and performed Co-IP assays to map HDAC2-binding domain in the SMAD3 protein. Our data show that the MH1 domain of SMAD3 is essential for binding to the HDAC association domain of HDAC2 protein (Please see Fig 4h-i; Supp Fig 14h-j). This repressive effect of SMAD3 on transcriptional activity of target genes has been shown to be dependent on its association with HDACs' enzymatic activity (Liberati et al, 2001). Furthermore, HDAC2 KO and SMAD3 KO BTSCs show decreased immunoprecipitation of HDAC2 protein with SMAD3 antibody, resulting in associated changes in cellular and molecular programs of BTSCs, further confirming the functional relevance of HDAC2 enzymatic activity in HDAC2-SMAD3 complex. The new data from Co-IP assays have been presented in Fig 6d and Fig 7c of the revised manuscript.
- 3) This is a very interesting point; unfortunately the only available antibody is not specific to acetyl-SMAD3 but detects the acetylated form of both SMAD2 and 3 (Catalog # PA5-76015, Invitrogen) and may conduce to providing any conclusive data to answer this question.

6. Technically, it is unclear how the authors conducted RNA-seq differential expression analysis – $\text{Log}_2(\text{fold change}) > 1.0$ or 1.5 ? How many genes are significantly up- or down-regulated (in Fig.1f)? Pathway and/or gene-set enrichment analyses are also missing.

We have now added these details in the results and method sections. We performed gene set enrichment analysis using the list of differentially expressed genes ($\text{FDR} < 0.05$) and the enriched gene sets were visualized using Cytoscape with enrichment map and autoannotate plug-ins. (Please see the Supp Fig 5).

7. Santacruzamate A (a selective HDAC2 inhibitor, PMID: 32839551) should be tested for verification of the romidepsin data.

We thank the reviewer for this suggestion and have screened Santacruzamate A in BTSCs lines and normal human fetal astrocytes. We found similar results as shown with romidepsin treatment. These findings support the HDAC2 KO data and confirm the high relevance of HDAC2 specific epigenetic regulations in BTSCs. (Please see the new data presented in the Fig 7f-i).

8. The results presented here do not fully support the working model (Fig.6e). For instance, no experimental evidence suggests SMAD4 as part of the HDAC2-SMAD3 complex. Moreover, it remains to be determined whether romidepsin influences SMAD3 phosphorylation.

To address this comment, we performed Co-IP using SMAD4 antibody and determined that its interaction with SMAD3 which was disrupted following treatment with romidepsin (Please see Supp Fig 14e). There was a decrease in the protein levels of both total and phospho-SMAD3 following romidepsin treatment and genetic deletion of HDAC2 that could be a due to transcriptional repression and NEDD4L mediated increase in the protein turnover of SMAD3 (Please see Fig 2b, 3b, c, Supp 8b, c). We have revised the working model Figure 8e to better reflect the key findings of this study.

Minor comments:

1. Page 1; Line 2 – Title: cells → cell.

This comment has been addressed.

2. Page 3; Lines 55-56 - aberrant methylation or acetylation histone patterns → aberrant histone methylation or acetylation patterns.

This comment has been addressed.

3. Page 5; Line 110 – HFAs → human fetal astrocytes (HFAs).

This comment has been addressed.

4. Page 7; Lines 151-152 – Supp Fig. 2a → 4a.

This comment has been addressed.

5. Page 7; Line 152 – patterns → levels.

This comment has been addressed.

6. Page 7; Lines 161-163 – In addition to western blot (Fig. 2a and Supp Fig. 4c), the authors should validate RNA-seq data by RT-qPCR.

To address this point, we performed RT-qPCR to validate the differential expression of genes such as SOX2, BDNF OLIG2 and SMAD3 following romidepsin treatment relative to the vehicle control samples. We observed similar changes in the mRNA abundance of these genes as those reflected at their protein levels. (Please see the new data added in the Supp Fig 7a-d of the revised manuscript).

7. Fig. 4d – Compared to other IP experiments (Figs. 4a-c), why was no ID signal detected in HDAC1+2 IP?

We thank the reviewer for this point and would like to clarify that this was a sequential Co-IP. According to the protocol, the lysate was first incubated with the HDAC1 antibody and then the eluted proteins were resuspended in buffer and incubated with HDAC2 antibody. During the process, it is expected to observe some depletion and dilution of proteins in the sequential ID sample compared to single antibody ID. We have added the protocol in the methods section.

8. The authors should conduct SKI IP to confirm its interaction with HDAC1/2 and SMAD3.

As per the reviewer's suggestion, we performed Co-IP using SKI antibody. Unfortunately, the available SKI antibody is not very efficient for Co-IP. However, we were able to see a band for SMAD3 and HDAC2, confirming the interaction between HDAC2-SMAD3 and SKI proteins in BTSCs. (Please see the new data from Co-IP assays in the revised Supp Fig 14d).

9. Supp Figs. 9a-b and 10a-b – Romidepsin treatment did not affect HDAC1 and HDAC2 protein expression in BTSC lines (Fig. 2a and Supp Fig.4c). But, why was the efficiency of HDAC1/2 IP under romidepsin treatment much lower than the vehicle control? This inconsistency might lead to an incorrect conclusion that romidepsin destabilizes the interaction between HDAC1/2 and SMAD3.

This is a good point from the reviewer. Based on our experimental findings we, explain these observations as follow:

1) In the co-IP assays following time course treatment with romidepsin, we observed that IP efficiency is impacted at 72 hours post-treatment. This could either be due to an impact on the viability of BTSCs or disruption of protein complex that impacting the crosslinking efficiency relative to vehicle control (Please see Supp Fig 14e).
2) Furthermore, stable KO of SMAD3 showed a slight reduction in HDAC2 protein which was also reflected in the levels of immunoprecipitated HDAC2 relative to the AAVS1 control BTSCs. These findings suggest that SMAD3 may have an impact in regulating HDAC2 expression which was evident in stable SMAD3 KO BTSCs than following romidepsin treatment. The new western blotting and Co-IP assays data from the SMAD3 KO and AAVS1 control BTSCs have been added in the Fig 7a, c and Supp Fig 19a of the revised manuscript.

10. Supp Figs. 9d and 10d – To fully validate the IP data, PLA experiments should be conducted in both vehicle and romidepsin-treated BTSCs.

This is a good suggestion from the reviewer, which helps in overcoming the technical drawbacks of Co-IP experiments. We performed PLA assays in romidepsin treated vs DMSO treated BTSCs and observed a significant decrease in total number PLA foci for samples probed with HDAC1-HDAC2, HDAC2-SMAD3 and HDAC2-SKI antibodies, further confirming the protein interactions between HDAC2-SMAD3 and SKI in BTSCs (Please see the revised data in the Fig 4g and Supp Fig 15a).

11. Fig. 4e – The cell line name was missing.

We apologize for this oversight; the cell line name has been added.

12. In ChIP-seq experiments, did HDAC2 and SMAD3 co-occupy the SOX2 locus? Since SOX2 is a key target of SMAD3, this finding should be validated by HDAC2/SMAD3 ChIP-qPCR and recruited to the main figure.

This is a very important point and our new CHIP-seq data, performed for the revisions, show more binding of SMAD3 than of HDAC2 at SOX2 gene locus. This confirms the findings from SMAD3 KO and rescue experiments, where KO or overexpression of SMAD3 altered the protein levels of SOX2, accordingly. (Please see the revised Fig 5c, h, 6a, 7a, Supp Fig 18a and 19a).

13. Page 14; Lines 328-329 – Compared to HDAC2-/Neg+, the median survival seemed to be reduced in the HDAC2-/SMAD3+ mice (Fig. 5n).

This comment has been addressed.

14. Would SMAD3 depletion affect the viability of BTSCs and the median survival of xenografted mice?

This is an interesting point and we have added new data showing that the KO of SMAD3 in BTSCs resulted in reduced expression of stem cell markers, SOX2 and OLIG2. There was also an improvement in overall survival in mice orthotopically xenografted with SMAD3 KO BTSCs relative to AAVS1 controls. The new median survival data from SMAD3 KO relative to the AAVS1 control BTSCs have been added in the Fig 7e and Supp Fig 19b).

15. NSCs express HDAC2 at a level similar to that seen in BTSCs (Fig. 1a). In this regard, what is the rationale for using NSCs (but not HDAC2-negative/low astrocytes) in HDAC2 overexpression experiments (Figs. 6a-d)?

This is a valid point however, the astrocytes derived from hiPSCs were resistant to transduction (only 10% transduction efficiency was achieved). So, alternatively, as a normal cell control, we used NSCs to overexpress HDAC2 to determine its contributions to the regulation of BTSC stemness and growth characteristics.

Reviewer #3, expertise in TGF- β signalling pathway/CSCs and GBM (Remarks to the Author):

Bahia et al present solid observations about the key role of HDAC2 in regulating brain tumour stem cells characteristics by performing in vitro and in vivo experiments as well as elucidating partially its mechanism of action. They propose that HDAC2 cooperates with SMAD3-SKI to maintain the tumorigenic potential in brain tumour stem cells. Inhibition of genetic manipulation of HDAC1/2 shows that HDAC2 has a more relevant role in regulating cell viability, self-renewal and differentiation by modulating chromatin accessibility. This work is of relevance to the field in order to understand why targeting the TGF-beta pathway in glioblastoma has not resulted in a better outcome for patients, this article indicates that epigenetic changes modulated by HDAC2 could influence TGF-beta pathway. Nonetheless, several important aspects of the manuscript need attention.

Major points

- Some of the western blots show clear results, others are not so clear; it will make it less speculative to show quantification of at least 3 western blots for all the immunoblots panels in the article (at least for Fig 2a-b, Fig 3a-c, Fig 5k, Fig 6b, Sup. Fig 4c,d,f, Sup. Fig 6a-c, Sup. Fig 12a,

We thank the reviewer for the comment and to address this point, we have added additional supplementary data showing the quantification of replicates of immunoblots for suggested figures. (Please see the new Supp Fig 6, 12, 18c, d, 20b-e, and 21c-e).

- The authors validate by WB some of the differentially expressed genes found in the RNASeq analysis, such as SOX2, BDNF and GFAP. It would be interesting to check other stem cell fate regulators such as Olig1 and 2.

We have added data showing an impact of HDAC1/2 or SMAD3 inhibition on stem cell marker, OLIG2. The new data have been incorporated in the revised Fig 3a-c, 7a, h, i and Supp Fig 8a-c, 19a.

- The authors claim that the modulation in the components of the TGF- β pathway was specific to the inhibition of HDAC1/2 with romidepsin and was not observed with any other HDAC inhibitors; however in Supp Fig 4f they only check Smad3 and Smad7; they should also check P-Smad3, NEDD4L and SKI.

To address this point, we treated the cells with pan- and specific-HDAC inhibitors including Santacruzamate A and checked the impact on stem cell, neuronal and TGF- β pathway related genes including p-SMAD3, NEDD4L and SKI. Specific inhibition of HDAC2 shows similar changes in these genes as seen following romidepsin treatment and in genetic KO studies. Please see the new Supp Fig 20.

- The authors claim that the modulation in the components of the TGF- β pathway is mediated by HDAC2; however in Fig 3a and Supp Fig 6a they don't check Smad7; P-Smad3, and NEDD4L which were initially also shown to be modulated by HDAC1/2 inhibitors, and it would be important to check that those effects are not unspecific of the inhibitors.

To address this point, we used HDAC1 and 2 KO BTSCs and checked the expression of SMAD7, p-SMAD3 and NEDD4L. Our data confirms that the modulation of these genes is specific to HDAC2. Furthermore, treatment with HDAC2 specific inhibitor, Santacruzamate A, confirms these findings. (Please see the new data presented in revised Fig 3a-c, Supp Fig 8a-c and Supp Fig 20).

- In order to show the importance of Smad3 loss upon HDAC2 KO, the authors rescue Smad3 levels by overexpression, showing that this rescues SOX2 levels and reduces GFAP and BDNF levels. It would be good to check here if Smad3 overexpression can also rescue the reduction of SKI observed upon HDAC2 KO shown in Fig3b-c.

This is a very interesting point and, as further investigation, we checked the expression of SKI following SMAD3 overexpression in HDAC2 KO BTSCs. We didn't observe an evident recovery of SKI expression. We propose that loss of SKI may result in lack of recruitment of HDAC2 to the SMAD3 protein complex, which thus may be responsible for this partial rescue of cellular and molecular properties of BTSCs (Please see revised Fig 6a and Supp Fig 18a).

- The authors could also check GFAP and BDNF expression in B147 after Smad3 overexpression (Sup fig 12a) to complement their findings in BT67.

This point has been addressed and new data has been added showing similar results in BT147 as observed in BT67 cells (Please see revised Supp Fig 18a).

- Smad3 overexpression partially rescues the effects induced by loss of HDAC2. Is it because it does not rescue SKI? If so, would SKI overexpression together with Smad3 completely rescue from the effects induced by loss of HDAC2 in cell viability, cell cycle, self-renewal and differentiation?

This is an interesting question and is unfortunately beyond the scope of this study to address. However, based on the rescue experiment, we speculate that exogenous expression of both HDAC2 and SKI may induce recover the HDAC2-SMAD3/SKI protein complex resulting in a complete rescue of cellular and molecular phenotype of BTSCs.

Minor points

- Line 152: Should be Sup Fig 4a

This comment has been addressed in the main document.

- Line 158-159: please add references that show that those genes are involved in neurogenesis, synaptogenesis and neural stem cell fates (it would be useful for people who are new to the field)

This comment has been addressed in the main document.

- Line 326: I believe AAVS1/SMAD3+ is missing from this sentence

This comment has been addressed in the main document.

- Line 404 mentions several references that should talk about TGF-beta effects on GBM BTSCs, however some of them do not refer to this topic.

We have revised the main document as per this suggestion.

- Line 408: there is no reference 65 in the reference list. Please double check the references cited

We have revised the main document as per this suggestion.

- General comment for figure legends: please add the time of treatment for each assay (cell viability, limiting dilution assay, Western blot, etc) it will be easier for the reader than having to go back and forth to the method section.

We have revised the figure legends as per this suggestion.

- Fig 2 legend says phospho-p38 while the figure shows p38 and there is no phospho-p38 antibody listed in the data set file.

We apologize for this error and have edited the correct information about the antibody used in this study.

- Fig3j: in the figure legend is mentioned **** while the graph shows only 3. They should match

We have edited the mistake with correct statistical information in the figure legend.

- Fig 5 legend caption: SKI levels are not checked; therefore, one can speculate that SKI is involved, but cannot assume. I think it would be better to remove SKI from the figure legend title. OR better, as mentioned in major points check SKI levels upon SMAD3 OE. And overexpress SKI together with S3 to see if overexpression of D3 and SKI together completely reverse HDAC2 KO effects in panel k, l and m

We agree with reviewers' point and to address this comment, we performed SKI IP and were able to show pull down of HDAC2 and SMAD3 (Please see Supp Fig 14d). We checked the SKI protein levels following SMAD3 overexpression in HDAC2 KO cells and found no obvious rescue of SKI expression (Please see Fig 6a and Supp Fig 18a). We speculate that in the absence of HDAC2 in HDAC2 KO cells, overexpression of both SMAD3 and SKI may not fully recover the formation of HDAC2-SMAD3/SKI repressive complex, unless they rescue the expression of HDAC2 protein, in order to rescue the HDAC2 induced changes in the chromatin structure and functions of BTSCs.

Reviewers' Comments:

Reviewer #1:

Remarks to the Author:

The authors have satisfactorily addressed all the comments and critiques with additional experiments. One comment that the authors should incorporate in the manuscript while interpreting the results with pharmacological agents like HDAC inhibitors is that these agents have higher potency for particular isoforms, e.g., Santacruzamate will have higher potency for HDAC2 but it is not specific for HDAC2 over other Class I HDACs like HDAC1. I would suggest the authors rely on the data from genetic knockout/knockdown for those conclusions. However, the manuscript is further improved and is sufficient for consideration in Nature Communications.

Reviewer #2:

Remarks to the Author:

The authors have thoroughly addressed my prior concerns with a considerable amount of new experiments. There are no further questions except for a couple of typos:

1) Figure 6a / Lane 4 – mislabeling: “SMAD3+” → “HDAC2-/SMAD3+”

2) The mean value of a “control” group needs to be set to 1.0 in the following bar charts:

- Figure 3e (AAVS1 control)
- Figure 7d (AAVS1 control)
- Supp Figure 7a-d (DMSO control)

Reviewer #3:

Remarks to the Author:

The authors have addressed all my comments.

The article is of high significance for the GBM field and TGF-beta field, showing a link between the epigenetic regulator HDAC2 and SMAD3 to maintain stemness in brain tumor stem cells. Therefore, I believe the publication of this article will be highly valued by the GBM and TGF-beta field.

We thank all the reviewers for providing positive feedback leading to acceptance of our manuscript.

REVIEWERS' COMMENTS

Reviewer #1 (Remarks to the Author):

The authors have satisfactorily addressed all the comments and critiques with additional experiments. One comment that the authors should incorporate in the manuscript while interpreting the results with pharmacological agents like HDAC inhibitors is that these agents have higher potency for particular isoforms, e.g., Santacruzamate will have higher potency for HDAC2 but it is not specific for HDAC2 over other Class I HDACs like HDAC1. I would suggest the authors rely on the data from genetic knockout/knockdown for those conclusions. However, the manuscript is further improved and is sufficient for consideration in Nature Communications.

We thank the reviewer for making this point. We have revised the interpretation of results from the section showing data from Santacruzamate A inhibitor as per reviewer's suggestion.

Reviewer #2 (Remarks to the Author):

The authors have thoroughly addressed my prior concerns with a considerable amount of new experiments. There are no further questions except for a couple of typos:

1) Figure 6a / Lane 4 – mislabeling: “SMAD3+” → “HDAC2-/SMAD3+”

We apologize for this error, and it has been corrected in the revised Figure6.

2) The mean value of a “control” group needs to be set to 1.0 in the following bar charts:
- Figure 3e (AAVS1 control)
- Figure 7d (AAVS1 control)
- Supp Figure 7a-d (DMSO control)

We thank the reviewer for these points. For the Figures panels pointed by the reviewer, the data have been normalized to the untreated controls to account for any off-target effects of vehicle control/gene editing procedure/selection methods. The y-axis legends reflect that the data have been normalized to the untreated controls.

Reviewer #3 (Remarks to the Author):

The authors have addressed all my comments.

The article is of high significance for the GBM field and TGF-beta field, showing a link between the epigenetic regulator HDAC2 and SMAD3 to maintain stemness in brain

tumor stem cells. Therefore, I believe the publication of this article will be highly valued by the GBM and TGF-beta field.

We thank the reviewer for the endorsement of our work and are looking forward to having the manuscript published.